# Multiscale interface engineering enables strong and water resistant wood bonding

Shiying Zhang[1], Salla Koskela [1,2], Halvar Meinhard [1], Paavo Penttilä [1], Muhammad Awais [1,3], Markus B. Linder [1,2], Shennan Wang [1] ✉ & Lauri Rautkari [1] ✉

The growing use of timber in construction has created an urgent need for high-performing engineered wood. Laminating timber facilitates production of structural components, but strong interfacial bonding is essential for engineered wood to outperform solid wood. Here we introduce a method for achieving strong wood bonding using an ionic liquid-dissolved cellulose solution. At the bonding interface, the dissolved cellulose fills the lumina and entangles with the wood cell wall, forming a dense cellulose network interconnecting with wood upon regeneration in water. Concurrent hot-pressing forms a permanently interlocked structure of wood cells. The multiscale bonded interface is water resistant with a shear strength over 20 MPa, nearly twice that of solid wood. This work presents an eco-friendly, high-performing wood bonding mechanism with promising applications in engineered wood products.

Increasing demand for sustainable buildings and new carbon emission mitigation techniques have shifted attention in sustainable materials science toward engineered wood. Engineered wood has a low carbon footprint, and it serves as an effective carbon storage during the lifetime of the material[1,2]. Mass timber, created by laminating wood layers, has already enabled the fabrication of large structural components with high mechanical performance, providing a sustainable alternative for concrete and steel in the construction industry[3]. Indeed, global urban development has been increasingly transitioning toward timber, and the current projections predict that by 2050, timber-based buildings could store up to 20 gigatons of carbon[4,5]. However, current fabrication techniques for mass timber still rely heavily on petroleum-based adhesives. In addition, many of the adhesives contain toxic precursors, such as phenolics and formaldehyde[6]. Therefore, there exists a pressing need for the development of high-performance, bio-based wood bonding methods to further enhance both the mechanical performance and environmental sustainability of mass timber.

Wood bonding with adhesives occurs mainly by flowing the adhesive into macroscopic cavities of wood, such as cell lumina, pits, and resin canals, forming an interconnecting network between the polymer and the hierarchical structure of wood[7]. Alternatively, friction welding can be used to facilitate self-adhesion of wood where mechanical energy is applied to melt the amorphous wood polymers such as hemicellulose and lignin, allowing them to flow and fuse into a dense structure at the bonding interface[8]. However, neither method ensures the bonded wood to surpass the shear strength of solid wood. In addition, the friction-welded wood is susceptible to swelling and disassembly when exposed to moisture, rendering it impractical for applications. The strength of the adhesive-mediated bonding is primarily influenced by the mechanical properties of the adhesive and its interfacial compatibility with wood cell wall components. For instance, the elastic modulus ($E$) of formaldehyde-based adhesive used in wood assemblies ranges from 0.1–15 GPa[9], whereas the wood cell wall possesses significantly higher $E$, up to 30 GPa, owing to the presence of strong, stiff, and naturally aligned cellulose microfibrils[10]. In addition, the potential interactions between the adhesive and the wood cell wall with a dense structure are often limited by poor mutual penetration. Consequently, the bonding line resembling a polymer-wood composite exhibits inferior theoretical mechanical performance compared to

[1]Department of Bioproducts and Biosystems, School of Chemical Engineering, Aalto University, Espoo, Finland. [2]Center of Excellence in Life-Inspired Hybrid Materials, Aalto University, Espoo, Finland. [3]Faculty of Science and Technology, Norwegian University of Life Sciences, Oslo, Ås, Norway. ✉e-mail: shennan.wang@aalto.fi; lauri.rautkari@aalto.fi

the intact wood cell wall. Friction welding enables densification and interlocking of wood cell walls, potentially increasing the volume fraction of cell walls at the bonding interface and creating strong bonding. However, it suffers from the formation of defects at the bonding line, primarily due to the insufficient flow of the melted cell wall polymers[11]. High temperatures (320–450 °C) generated from the intensive friction (100–150 Hz) also cause thermal decomposition of the wood cell wall components, especially hemicellulose, further compromising the bonding performance[12].

Herein, we introduce a wood bonding strategy that creates a robust bonding interface and achieves an extraordinary bonding performance. This approach takes advantage of the remarkable solubility of both cellulose and native wood cell wall in a room-temperature ionic liquid (IL)−1-ethyl-3-methylimidazolium acetate [emim][OAc]. Swelling of the wood cell wall and introduction of adhesive polymers to the bonding line are spontaneously achieved by using cellulose-IL solution as a bonding agent. The softened cell walls are then thermomechanically densified to form an interlocked structure, which

becomes permanently set by regenerating the dissolved cellulose and cell wall polymers in water. Our method enables the formation of a multiscale bonding interface that consists of micrometer-scale interlocked wood cell walls, an interconnecting regenerated cellulose network, and a highly compatibilized interface between regenerated cellulose and the wood cell wall. The resulting interface demonstrates superior mechanical strength and water resistance, outperforming many of the existing bonding methods. In addition, we have comprehensively studied the cellulose and wood cell wall regeneration mechanisms, along with a detailed investigation of the structure and properties of the bonding line in wood assemblies.

## Results

### Wood bonding with cellulose-ionic liquid solution

Freeze-dried bleached kraft pulp (viscosity-average degree of polymerization, $DP_v = 2360$) was dissolved in [emim][OAc] at a concentration of 5 wt.% to form a yellowish and transparent IL solution of pulp (Fig. 1a). Scots pine (*Pinus sylvestris*), with an oven-dry density of

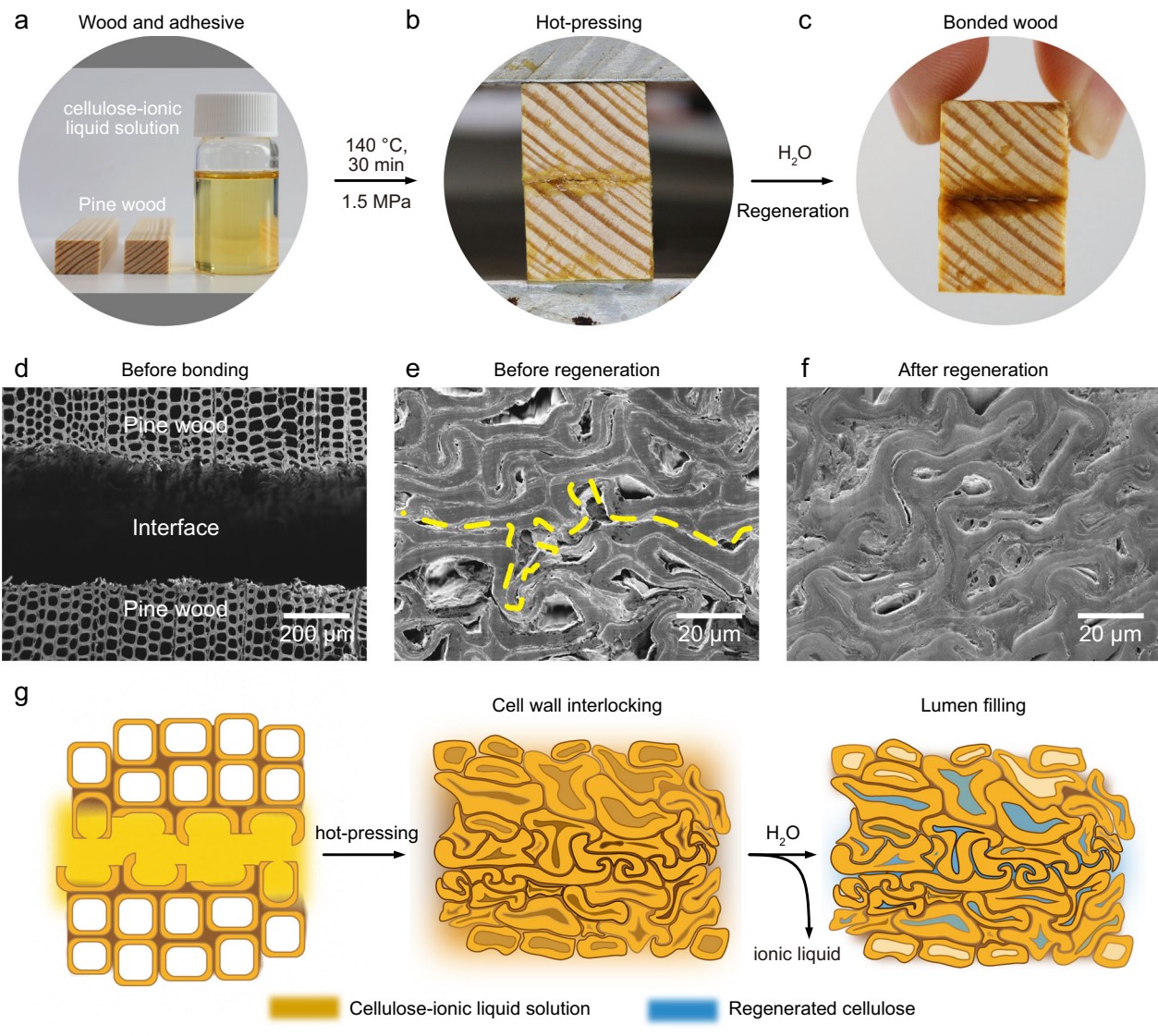

**Fig. 1 | Wood bonding with cellulose-ionic liquid (IL) solution.** Photographs of (**a**) pine wood and the cellulose-IL solution used as the bonding agent, (**b**) wood bonding by hot pressing, and (**c**) bonded wood after hot pressing and rinsing in water. Scanning electron microscopy (SEM) micrographs of (**d**) bonding interface between two pieces of wood, (**e**) bonding interface after hot-pressing before

regeneration (yellow dashed line indicates the boundary between two wood substrates), and (**f**) bonding interface after hot-pressing and rinsing in water.
**g** Schematic illustration of the microstructural changes during the wood bonding process with cellulose-IL solution. The cellulose-IL solution is indicated in yellow, and regenerated cellulose is depicted in blue.

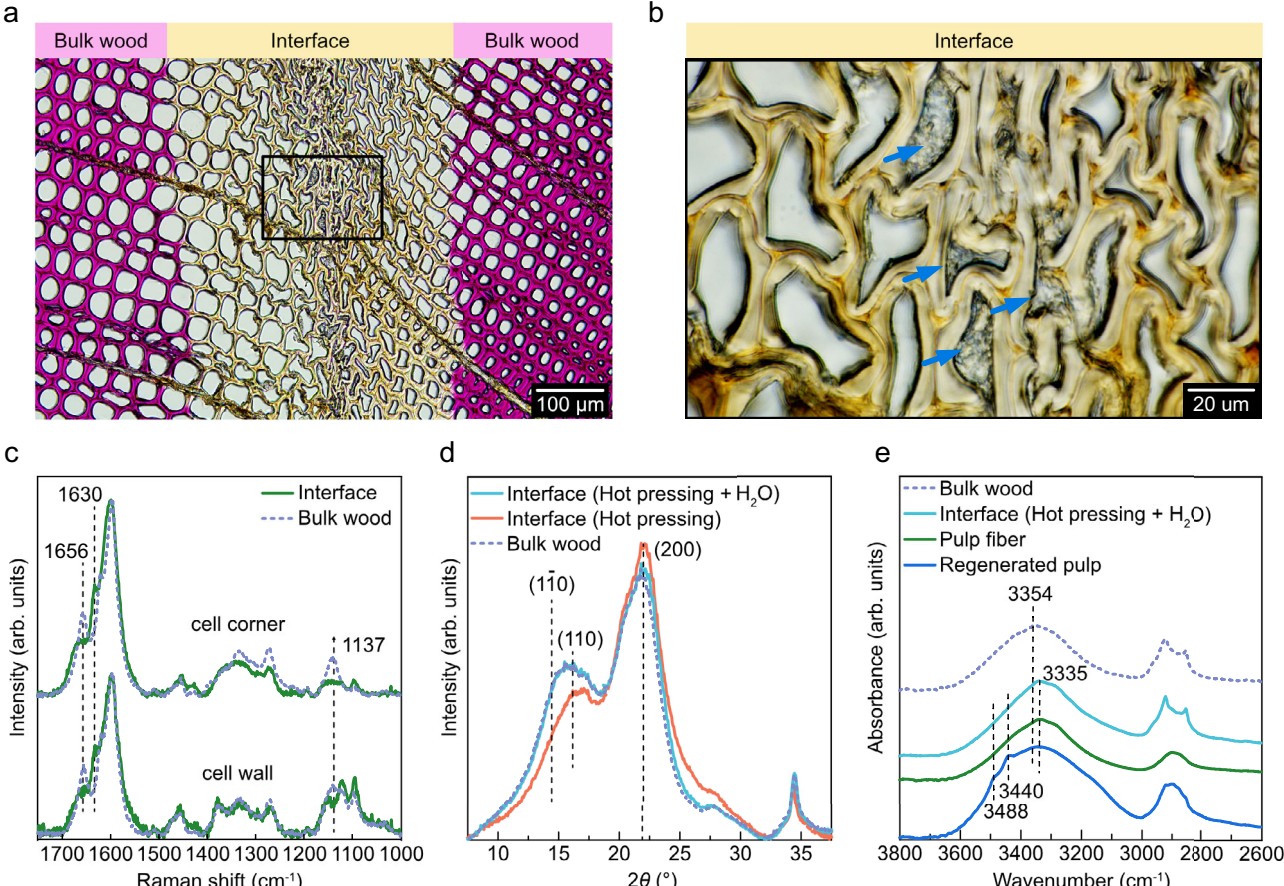

**Fig. 2 | Structure and composition of the wood bonding interface. a** Light microscopy image of a thin section of the bonded wood in water-saturated state, where the presence of native lignin is visualized by Wiesner staining (magenta). Light yellow highlights the wood bonding interface. **b** Light microscopy image at higher magnification showing the structure of the bonding interface with blue arrows indicating regenerated cellulose in the cell lumina. **c** Raman spectra of the bonded wood sample showing the signals acquired from cell wall and cell corner regions at the bonding interface (green solid lines) and the bulk wood (purple dashed lines). **d** Wide-angle X-ray scattering (WAXS) patterns of bulk wood and bonded interfaces before (orange) and after (light blue) rinsing in water. **e** Attenuated Total Reflectance-Fourier Transform Infrared (ATR-FTIR) spectra of bonded wood (purple), regenerated pulp (blue), prepared on a glass slide, and original pulp fiber (green).

480 kg m$^{-3}$, was used as adherend for the bonding test. First, pulp-IL solution was homogenously applied to the adherends' interface at a solid spread rate of 9 g m$^{-2}$, followed by hot-pressing at 1.5 MPa and 140 °C for 30 min (Fig. 1b) to promote interactions between wood and pulp-IL solution. The assembled wood was then rinsed with deionized water to remove IL and to facilitate regeneration of the dissolved pulp at the bonding interface. In the bonded wood, the two adherends become fused (Fig. 1c), showing a brownish seam at the junction site, indicating successful bonding.

Scanning electron microscopy (SEM) micrographs obtained from transverse sections of the wood specimens during the bonding process (Fig. 1d–f) revealed the microstructural changes in the wood cellular structure at the bonding interfaces. Before bonding, the pine wood adherends displayed typical wood cellular structures (Fig. 1d). During hot pressing, heat and pulp-IL were found to facilitate densification of the wood cell wall (Fig. 1e). Notably, the cell walls of both adherends intertwined, forming a mechanically interlocking structure[8]. After the regeneration of the dissolved pulp in water (Fig. 1f), the un-densified cell lumina and cavities that were still visible in Fig. 1e were now found to be filled in, most likely by regenerated cellulose. Remarkably, at the micrometer scale, there was no visible adhesive layer typically observed in adhesive-bonded wood. On the contrary, the bonding interface became indistinguishable from the bulk material, suggesting that heat and the pulp-IL solution effectively

promoted cell wall fusion. At higher magnification, regenerated cellulose was found seamlessly attached to the lumen surfaces, indicating good interfacial interactions between the lumen filling polymer (regenerated cellulose) and the wood cell wall (Supplementary Fig. 1). Figure 1g further illustrates the bonding mechanism at the interface.

## Molecular-level structural changes at the bonding line

To further investigate the composition of the bonding line, we applied Wiesner staining (phloroglucinol) to a 20 µm-thick section of the bonded wood. Wiesner staining enables visualization of qualitative structural changes of lignin that might have occurred due to contact with the pulp-IL solution. The light microscopy images in Fig. 2a and Supplementary Fig. 2 revealed that the tracheids in the bulk wood region were readily stained in magenta color. This phenomenon results from the characteristic reaction between phloroglucinol and coniferyl aldehyde in lignin[13,14]. However, the tracheids around the bonding interface were not stained, indicating reduction of the coniferyl aldehyde content in this region, which was further supported by Raman spectroscopy (Fig. 2c). At the bonding interface, the intensity of the peaks at 1656 and 1137 cm$^{-1}$ corresponding to the C=O and C−C stretching vibrations in coniferyl aldehyde[15] were significantly reduced as compared to the bulk wood. The reduced intensity of these peaks, both at the cell wall and cell corner regions, was further accompanied by the emergence of a new peak at 1630 cm$^{-1}$, attributed to the C=C

stretching in stilbene structure. These results suggest that the wood cell wall lignin became partially dissolved in the pulp-IL solution, resulting in conversion of coniferyl aldehyde to stilbene, a phenomenon previously observed for solid wood modification using ionic liquid 1-ethyl-3-methylimidazolium chloride[16,17].

We also performed wide-angle X-ray scattering (WAXS) to investigate the structure of the cellulose microfibrils at the bonding interface (Fig. 2d). Bulk wood showed a typical scattering pattern of cellulose $I_\beta$, with three main scattering peaks centered at $2\theta = 14.6°$, 16.2°, and 21.7°, corresponding to the (1−10), (110), and (200) lattice planes, respectively. Directly after application of pulp-IL and hot pressing, the peak heights at $2\theta = 14.6°$ and 16.2° were observed to decrease at the bonding interface, while the relative intensity of the peak at $2\theta = 21.7°$ was enhanced. The selective disruption of (1−10) and (110) surfaces of cellulose $I_\beta$ crystallites indicates the weakening of intermolecular interaction between cellulose chains in the crystalline region by pulp-IL solution, resulting in swelling of the wood cell wall. The swelling of wood cell walls was confirmed through light microscopy by incubating a thin slice of solid wood in the pulp-IL solution (Supplementary Fig. 3). Initially, the thickness of compound cell walls of tracheids measured approximately 6.1 µm in earlywood regions and 10.6 µm in latewood regions (Supplementary Table 1), with negligible changes observed after incubating at 20 °C for 60 min. However, earlywood tracheids with a typical rectangular shape transformed to a rounded shape after 10 min incubation at 60 °C, with cell wall thickness doubling to 12.6 µm. As a result of this swelling, the cell wall thickness increased more than twofold after 30 min of incubation at 60 °C, while the space within lumina became minimized. After rinsing the wood slice with water, the swollen cell walls contracted to an intermediate state, with approximately 60 % thicker cell walls than their initial thickness, indicating structural changes induced by the water-triggered regeneration process. Similar phenomena were also observed for latewood tracheids. The WAXS pattern of the bonded interface after rinsing with water showed almost identical intensity at $2\theta = 14.6°$ and 16.2°, as compared to bulk wood (Fig. 2d). This phenomenon can be explained by recrystallization of previously disrupted cellulose microfibrils, which has also been observed on ionic liquid treated microcrystalline cellulose (MCC)[18] and ionic liquid treated densified wood[19]. None of the measured WAXS patterns showed any detectable amount of cellulose II crystals, even though they could be expected to form as a consequence of cellulose regeneration. In addition, when comparing the Fourier transform infrared (FTIR) spectra of the samples (Fig. 2e), we observed a peak shift from 3354 cm$^{-1}$ in the bulk to 3335 cm$^{-1}$ at the interface. However, the molecular structure of the bonding interface remained consistent with the bulk wood in the 1800-600 cm$^{-1}$ range (Supplementary Fig. 4). Considering the FTIR spectrum of the bonding interface exhibited substantial similarity to the pulp fiber within the 3600-3000 cm$^{-1}$ region, the peak shift at the interface is likely due to the regenerated cellulose from the pulp-IL solution. In addition, characteristic peaks of cellulose II at 3440 and 3488 cm$^{-1}$ [20], presented in the separately prepared regenerated pulp, were absent in the spectrum of the bonding interface. This implies that water-induced cellulose regeneration does not consistently result in a transition from cellulose I to cellulose II, aligning with the WAXS result.

## Distribution and cellular penetration pathway of the bonding agent

Determining the distribution of the regenerated cellulose at the bonding interface is important for understanding the penetration pathway of pulp-IL during the bonding process. It is also crucial for elucidation of the bonding mechanism. However, differentiation between the native cell wall cellulose and the introduced cellulose, from pulp, at the submicron scale is challenging due to their identical chemical structures. To overcome this, we employed cationic pulp-IL. Cationic pulp with detectable trimethylammonium chloride groups should enable distinction between the native cellulose from the cell wall and the regenerated cellulose, and determine the distribution of the latter. Although cationic modification of cellulose may alter its distribution and interactions with lignin, this approach provides valuable spatial and chemical insights into the formation of the bonding interface. First, the cationic pulp was prepared from bleached kraft pulp using glycidyl trimethylammonium chloride (GTMAC) in alkaline conditions (Fig. 3a). After the covalent modification, partial substitution of hydroxyl groups on cellulose with trimethylammonium chloride could be confirmed by their unique absorption band at 1478 cm$^{-1}$ in the FTIR spectrum (Fig. 3c)[21]. Sedimentation test showed that the original pulp started to precipitate after standing for 5 min, while the cationic pulp remained colloidally stable, indicating successful introduction of charged trimethylammonium chloride onto the cellulose microfibrils (Fig. 3b). The freeze-dried cationic pulp was subsequently dissolved in [emim][OAc] to a 5 wt.% concentration, which was then used as the wood bonding agent. Notably, the viscosity of the cationic pulp solution was akin to that of the non-modified pulp solution (Supplementary Fig. 5), and thus both solutions should exhibit comparable flow and bonding behavior.

Next, we utilized Raman mapping to track the trimethylammonium chloride group in the cationic cellulose. Due to the sensitivity of the confocal Raman spectroscopy, the spectrum of wood typically exhibits strong signal interference from lignin, as demonstrated in Supplementary Fig. 6. To mitigate this interference and to enhance the visibility of signals from the regenerated cationic cellulose, a thin transverse section with a thickness of 25 µm was prepared from the cationic pulp-bonded wood and partially delignified (Fig. 3d)[22]. Partial delignification was found to remove the brownish color of the bonding line by detaching any chromophores. As expected, the Raman spectrum obtained from the delignified bonding interface showed an intensified peak at 764 cm$^{-1}$ (Fig. 3e), corresponding to the C − N stretching band in the trimethylammonium group, along with a characteristic peak for cellulose II at 576 cm$^{-1}$ [23,24]. In addition, these peaks were not found in the Raman spectrum from the cell wall of bulk wood, supporting its use as a suitable indicator to detect the regenerated cellulose.

Raman color maps taken from the mixed bonding interfaces between earlywood or latewood are shown in Fig. 3f, with the color gradient indicating the relative intensity of the C − N stretching band from the trimethylammonium group. It was evident from the color maps that the regenerated cationic cellulose predominantly filled the lumina of earlywood tracheids at the bonding interfaces of earlywood-to-earlywood and earlywood-to-latewood. Its presence in the cell wall was also detected, though less pronounced than in the lumina. This penetration behavior resembles that of formaldehyde-based adhesives as well as other low viscosity adhesives, which typically flow through pits between tracheids or fiber cells to form a network of adhesive polymers enabling efficient bonding[25]. In latewood-to-latewood bonding interfaces, the regenerated cellulose did not pass to the lumen, but mostly stayed at the interface region.

However, some of the cationic cellulose was able to pass to the middle lamella and cell corner. The spectra extracted from these regions can be found in Supplementary Fig. 7. Therefore, the middle lamella was identified as a secondary penetration pathway for cationic pulp-IL solution during bonding, a phenomenon that has not been reported in conventional adhesive-based wood bonding. The middle lamella constitutes mostly of lignin, which serves as a binder between the tracheids. During hot pressing, lignin was presumably in a molten state as the hot-pressing temperature (140 °C) reached above the glass transition temperature of lignin[26]. Indeed, it is likely that the IL-dissolved cationic pulp penetrated into the middle lamella during this time.

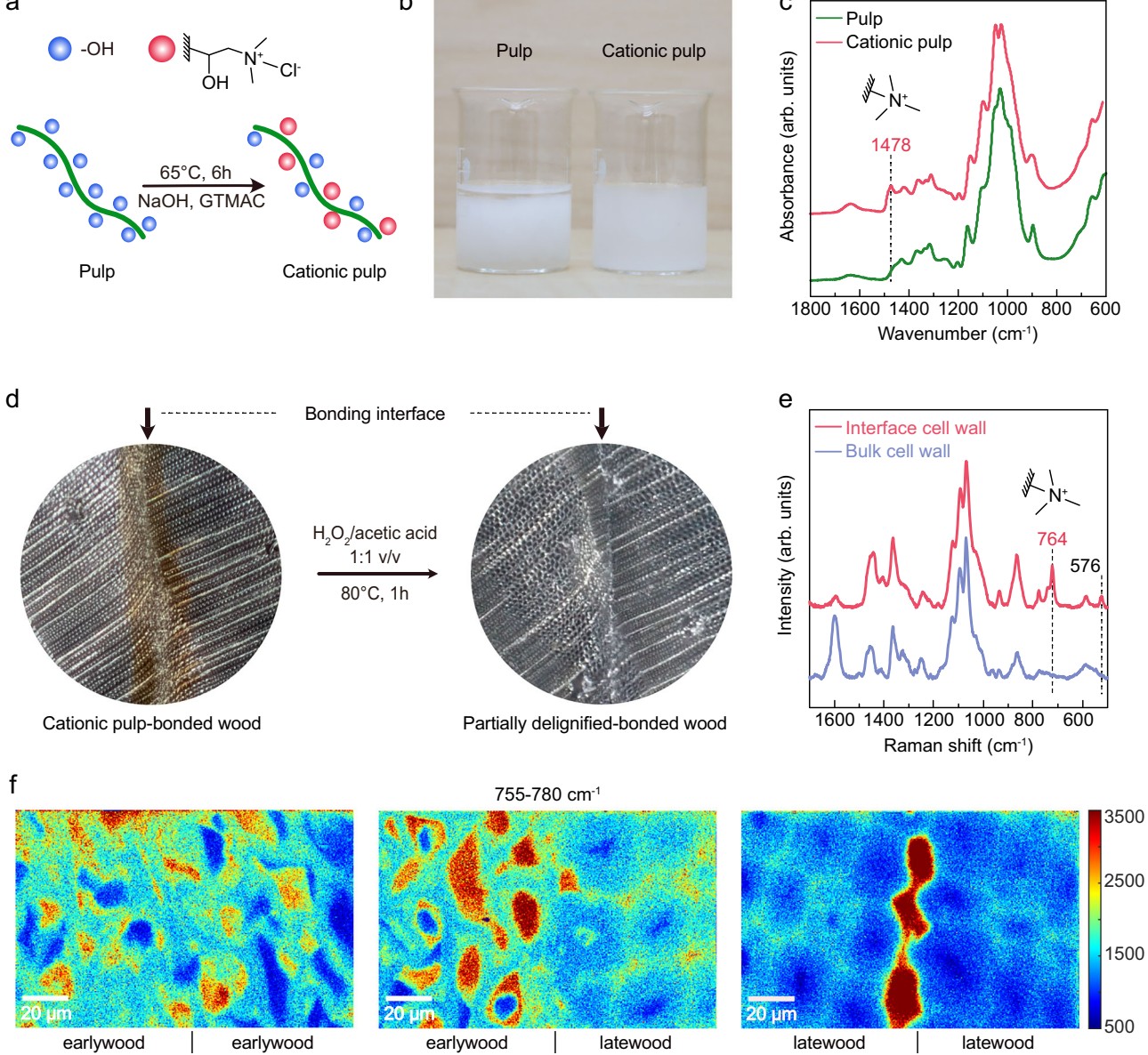

**Fig. 3 | Distribution of regenerated cellulose at the bonding interface.**
**a** Schematic illustration of the preparation of cationic pulp. The green curve represents pulp fiber. Surface hydroxyl groups (–OH, blue spheres) are partially substituted by trimethylammonium chloride functional groups (pink spheres). **b** Photograph of pulp and cationic pulp with a solid content of 2 wt.%. **c** ATR-FTIR spectra of pulp (green) and cationic pulp (pink). **d** Partial delignification of wood bonded with cationic pulp-ionic liquid solution. **e** Raman spectra from the cell wall of bulk wood (purple) and the lumen region at the bonding interface (pink). **f** Raman mapping on the transverse section of the bonding interface reveals the distribution of regenerated cationic cellulose between different types of wood tracheids. The colored map demonstrates the integration of Raman bands from 755–780 cm$^{-1}$, corresponding to the peak area of 764 cm$^{-1}$.

Surprisingly, cell lumina at the earlywood bonding interfaces that contained cationic cellulose were also found to contain lignin. The presence of lignin was observed as a characteristic band at 1600 cm$^{-1}$ in the Raman spectrum (Fig. 3e), attributed to the C = C stretching in the phenolic rings of lignin. It is probable that IL-dissolved cellulose and the molten lignin became physically entangled during the wood bonding process, resulting in mobility of lignin into the cell lumina, where cellulose was subsequently regenerated. The presence of lignin in the lumina was further proven by Raman mapping of lignin (Supplementary Fig. 8).

### Effect of cellulose molecular weight on formation of the bonding interface

Resembling the behavior of various thermoplastic adhesives, and other natural polymer-based adhesives such as locust bean gum[6] and chitosan[27], cellulose-IL solutions do not undergo polymerization during the wood bonding process. Therefore, we speculated that the initial molecular weight of cellulose would play a critical role in its penetration into the wood cellular structure and its ability to regenerate into a network. To test this hypothesis, in addition to pulp, we also investigated the use of two lower molecular weight celluloses, MCC and microfibrillated cellulose (MFC), to prepare wood bonding agents. The experimentally determined DP$_v$ of pulp, MFC, and MCC are 2360, 1470, and 180 (Fig. 4c), respectively, and thus the order of the cellulose molecular weight of these is pulp > MFC > MCC.

As shown in Fig. 4a, dry MCC is clearly different from the other two celluloses with a powder-like texture. By contrast, freeze-dried MFC and pulp fibers both display foam-like structures owing to their considerably higher aspect ratios. After being dissolved in [emim][OAc], all cellulose-IL solutions became yellow (Fig. 4b). Despite having

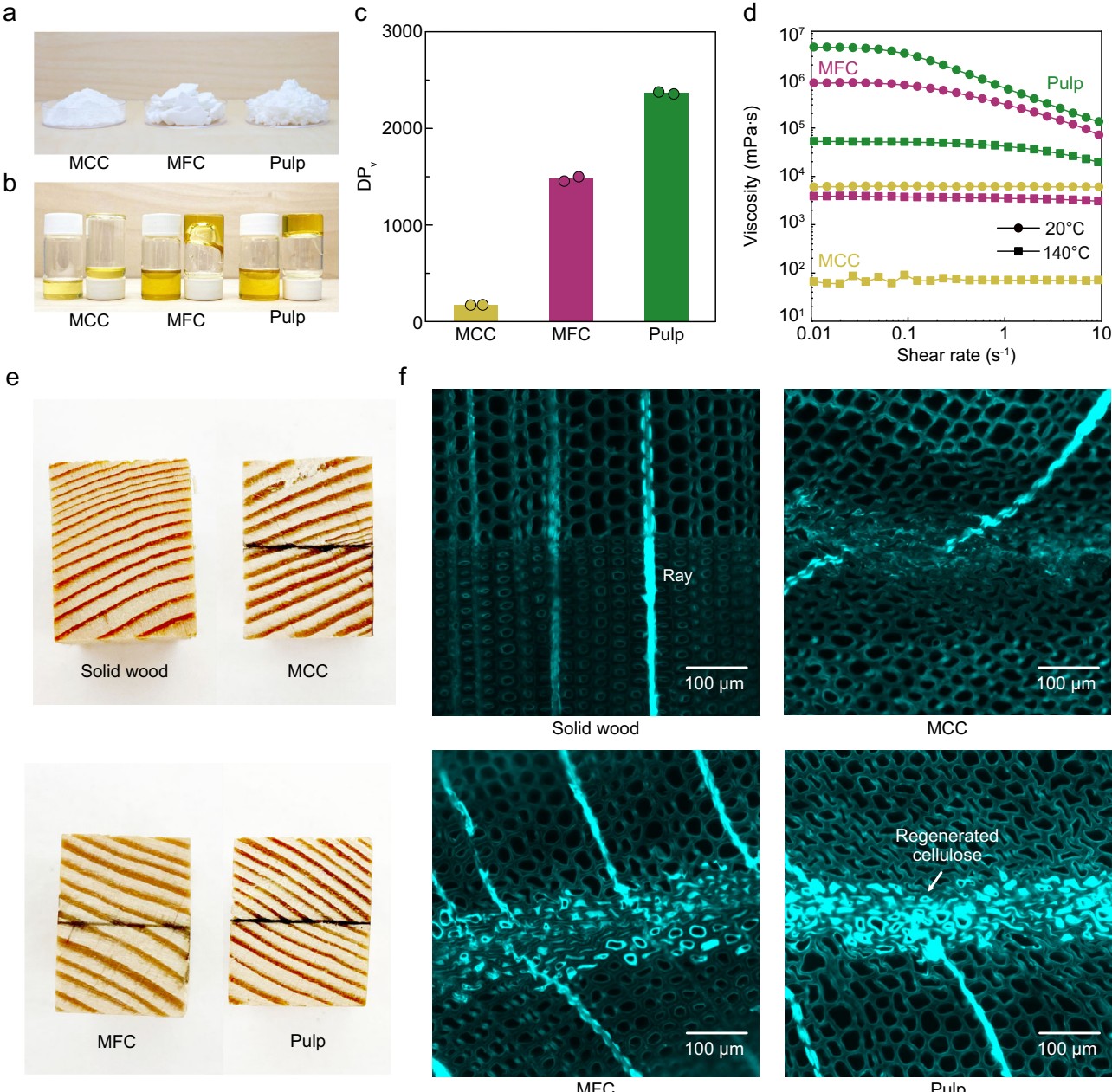

**Fig. 4 | Cellulose-ionic liquid solutions prepared from various types of cellulose for wood bonding. a** Photograph of microcrystalline cellulose (MCC), microfibrillated cellulose (MFC), and bleached kraft pulp. **b** Photograph of ionic liquid solutions of MCC, MFC, and pulp at a solid content of 5 wt.%. **c** Viscosity-average degree of polymerization (DP$_v$) of MCC, MFC, and pulp used for wood bonding. Data is presented as the mean of $n$ = 2 replicates. MCC, MFC, and pulp are represented in yellow, magenta, and green, respectively. **d** Representative flow curves for various cellulose-ionic liquid solutions of 5 wt.% solid content at 20 °C (dots) and 140 °C (squares). **e** The solid wood (control) and wood assemblies successfully bonded with MCC, MFC, and pulp, respectively. **f** Fluorescence microscopy images showing calcofluor white stained cross-sections of native (solid) wood, and those of wood bonded using various celluloses.

the same solid content (5 wt.%), DP$_v$ of cellulose was found to significantly influence the viscosity of the solutions. At room temperature, both the pulp and MFC solutions exhibited rather gel-like behavior, whereas the MCC solution behaved as a liquid (Fig. 4b). We investigated the flow behavior of cellulose-IL solutions at both 20 and 140 °C, i.e., the hot-pressing temperature. At both temperatures, Newtonian plateaus (0.01–0.1 s⁻¹) were recorded for all cellulose solutions. Their steady-state viscosities were found to decrease when the temperature was increased from 20 to 140 °C (Fig. 4d), indicating the enhanced flowability of cellulose solutions during hot-pressing. Next, we prepared bonded wood using the three different cellulose-IL solutions (Fig. 4e) as described before.

The behavior of the various cellulose-IL solutions to penetrate and regenerate in the wood cellular structures was studied with fluorescence microscopy. For this, we selected a fluorescent dye, calcofluor white. Having a high affinity for cellulose[28], calcofluor white does not stain pentosans from hemicellulose[29], or lignin[30]. Thus, it should enable us to visualize the regenerated cellulose located in the cell lumina. First, we stained thin transverse sections of the wood samples bonded with either MCC, MFC, or pulp. For a negative control, we similarly stained a thin section of native wood without any bonding treatment. The obtained corresponding fluorescence microscopy images of the solid wood, MCC, MFC, and pulp-bonded wood interfaces are shown in Fig. 4f.

The microscopy images revealed that although all the tested celluloses were able to facilitate wood bonding, the formed bonding interfaces were different. Evidenced by the presence of fluorescence from the calcofluor white-stained regenerated cellulose, MCC formed the thinnest bonding interface with an approximate thickness of 88 µm. By contrast, both MFC and pulp fibers formed much thicker bonding interfaces with an approximate thickness of 120 µm (Fig. 4e). Pulp fibers with high molecular weight were efficient in forming a visible matrix within the cell lumina, apparent from the very intense fluorescence from the cells at the bonding interface. That, and the fact that the shape of the wood cells stayed relatively intact despite the hot-pressing, indicated complete filling of the lumina with regenerated cellulose. MFC was able to fill the cell lumina in a rather similar manner, although the wood cells appeared slightly more compressed than those filled with regenerated pulp. Interestingly, MCC with the lowest cellulose molecular weight showed strikingly different behavior. The wood cells at the interface bonded with MCC appeared completely deformed, with a faint fluorescence signal from the regenerated MCC distributed within this deformed matrix. It should be noted that calcofluor white fluorescence was also observed in the ray cells (Fig. 4e). As the ray cells in solid wood without any modification were also stained, binding of calcofluor white to these cells was attributed to the low lignin content in their cell walls[30] and not to the presence of regenerated cellulose.

The differences in the formation of the bonding interfaces were attributed to the different cellulose molecular weights of the tested substrates and to the inherent structure of the wood cell wall. Wood cell wall decorated with pits most likely works as a sort of filter during the hot pressing. Pressure generated during the hot pressing pushes the polymer solution from the bonding surfaces through the pits that are interconnecting the wood cells in the transverse direction. Because of the relatively easier passage of ionic liquid as compared to the dissolved celluloses through this filter, the ionic liquid can penetrate further away from the bonding surface than the dissolved cellulose molecules. The penetration distance of the ionic liquid is best visualized in Fig. 2a, being roughly 450 µm from the bonding surface.

It appears that the molecular weight of the dissolved cellulose further affected their passage through the pits. In the case of MCC, the low molecular weight potentially enhanced its penetration through the pits, enabling it to distribute into more cells as compared to MFC and pulp celluloses. MFC and pulp cellulose, having much larger molecular weights, likely become jammed inside the cells near the bonding interface, where they become regenerated. Even as dissolved molecules, their presence in the lumen enabled the cell to retain some of its initial shape during hot pressing. Furthermore, it is likely that some of the more distributed MCC was washed away during rinsing in water, as it failed to generate a strong entangled network within the cells. Indeed, the results demonstrate that higher molecular weight cellulose enables the formation of a regenerated cellulose matrix inside the cells with possible physical entanglement with the cell wall polymers, effectively interconnecting adjacent tracheids at the bonding interface.

## Bonding strengths achieved by various cellulose-ionic liquid solutions

The bonding strengths achieved by the different cellulose-ionic liquid solutions were evaluated by tensile shear test on lap-joint wood samples (Fig. 5a, Supplementary Movie 1, and Supplementary Fig. 9). In addition, native solid pine wood and samples bonded using hot-pressing and MCC-IL without the regeneration step were measured as controls.

Upon the lap shear test, the solid wood sample fractured at a strain-to-failure of 3.8%. The measured shear strength of 12.2 MPa (Fig. 5b) was on par with previous measurements[31]. Compared to controls, all the wood assemblies showed improved bonding performance. The shear stiffness for pulp-IL and MFC-IL bonded wood were 496 and 508 MPa, respectively, higher than that of the solid wood (392 MPa). The highest shear stiffness (527 MPa) was observed for the MCC-IL bonded wood, likely due to the highest extent of densification at the bonding interface. Higher stiffness, together with enhanced strain-to-failure, resulted in excellent shear strengths of 15.7, 17.2, and 19.6 MPa for MCC, MFC, and pulp-bonded wood (Supplementary Table 2), respectively. Importantly, all the wood assemblies bonded with cellulose-IL solutions fractured adjacent to bulk wood rather than at the bonding interface (Fig. 5c and Supplementary Fig. 10). By contrast, the sample bonded without cellulose regeneration showed the lowest mechanical properties (11.9 MPa), fracturing directly at the bonding interface. SEM image of the fracture surface of pulp-IL bonded wood confirmed that the eventual fracture of the tracheids happened away from the bonding interface (Fig. 5d). In fact, the fracture site appeared to locate farther and farther from the bonding interface, when the DP of the cellulose used in the bonding increased. Similar fracture away from the bonding interface was also reported for high-performance wood-plastic composites when strong adhesion between the polymer and wood cell wall was achieved[32].

Rather surprisingly, the strain-to-failure of the MCC-IL bonded wood (4.2%) was close to that of the MFC-IL bonded wood (4.7%, Supplementary Table 2) despite the molecular weight of MCC being lower. This phenomenon is likely due to the fact that the sample bonded with MCC-IL was more densified at the bonding interface, as can be observed by the fluorescent images in Fig. 4f. Thus, the mechanical interlocking of the wood cells was enhanced in the case of MCC. This, together with the observation that the sample bonded by hot pressing without cellulose regeneration showed low mechanical properties, confirmed that both the formation of an interlocked structure and lumen filling with regenerated cellulose are important for enabling strong wood bonding. The combination of densified wood cell walls and regenerated cellulose likely allowed efficient stress transfer from the bonding interface, ultimately leading to bulk wood fracture at a higher stress level than the nominal shear strength of solid wood. A multiscale bonding mechanism is therefore revealed, benefitting from the mechanical interlocking of wood cells, the interconnecting regenerated cellulose network, and the enhanced interaction between the regenerated cellulose matrix and the wood cell wall.

Generally, the shear strength of adhesive-bonded and friction-welded wood shows dependence on the density of wood adherends and cannot ensure higher performance than solid wood (Fig. 5e). Importantly, in this work, all the wood samples bonded with regenerated cellulose surpassed the shear strength of solid pine wood. This enhancement was also observed for Norway spruce (*Picea Abies*) with a lower density of 320 kg m⁻³. A high shear strength of 14.0 MPa was achieved by bonding with pulp-IL (Supplementary Figs. 11 and 12). This is ideal, as the mechanical strength of the bonding line in engineered wood should exceed that of solid wood to ensure that the failure occurs through wood fracture rather than by failure at the adhesive layer[33]. It is also worth noting that the obtained values also exceed the minimum service requirement (10 MPa, EN 15425:2023) for bonded timber products. Although there have been attempts to use either ionic liquids[34] or cellulose solutions[35,36] for wood bonding, none of these studies successfully achieved the formation of an interlocked cell wall structure combined with a regenerated polymer network filling the cell lumina−features that are critical to the high bonding performance demonstrated in this work.

To further test the endurance of our bonded wood in harsh conditions, we conducted a delamination test for the pulp-IL bonded wood sample according to the EN 15425:2023 standard. The obtained results showed no bonding delamination after being submerged in boiling water for 6 h followed by soaking in 20 °C water for 2 h (Fig. 5f, Supplementary Movie 2, and Supplementary Fig. 13). This indicates

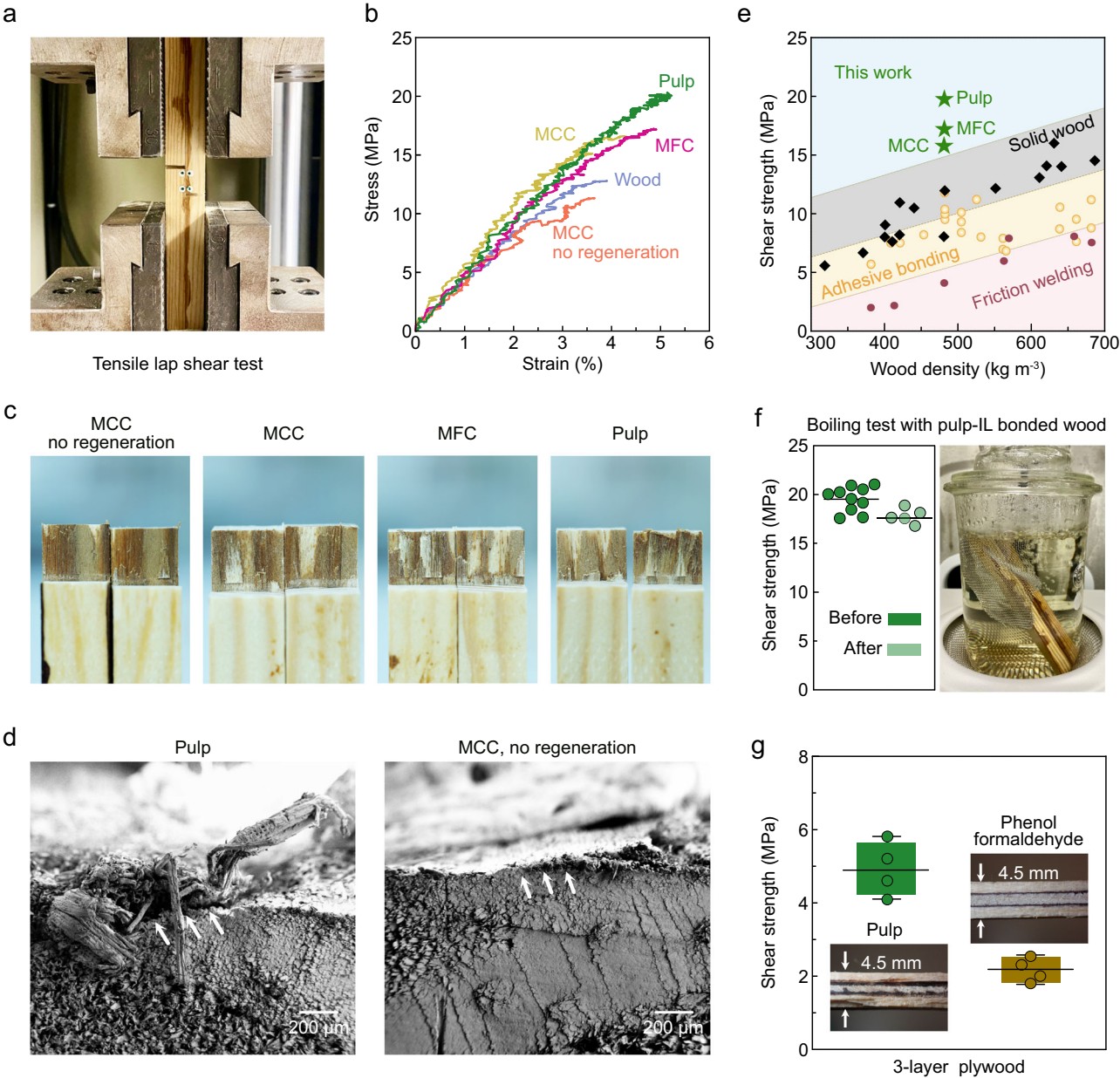

**Fig. 5 | Bonding performance of the wood assemblies. a** Photograph of the tensile shear test setup for the lap-joint wood specimens. **b** Representative stress-strain curves of tensile shear test of the wood samples bonded with different cellulose-ILs, using hot pressing in the presence of non-regenerated MCC (orange), and that of solid wood (purple). **c** A photograph of the fractured samples after the tensile shear test. **d** SEM micrographs of the fractured surfaces in bonded wood samples. White arrows indicate the location of the bonding interface. **e** A compilation figure showing the shear strength as a function of density for various wood adherends. The properties of solid wood, adhesive-bonded wood, friction-welded wood are

shown as black rhombus, beige circles, and dark red circles, respectively. The values obtained in this work are shown with green stars. The detailed information is listed in Supplementary Table 3. **f** Photo of boiling test of the pulp-bonded wood and the mean dry shear strength of the bonding line before ($n = 10$, green dots) and after ($n = 5$, light green dots) boiling test. **g** Photographs of cross sections of 3-layer plywood specimens of birch veneers bonded with pulp-IL (green) and commercial phenol formaldehyde resin (dark yellow), along with their shear strength. Data are presented as mean ± standard deviation (s.d.) of $n = 4$ replicates.

that the bonding interface has excellent water resistance. A lap shear test performed on the boiled samples showed shear strength of 17.8 MPa (Supplementary Fig. 14), retaining 90% of the performance of freshly bonded wood.

Moreover, the applicability of our method for a broad range of wood assembly products was demonstrated by a successful lamination of birch veneers. Using the pulp-IL, we were able to assemble a 3-layer plywood (Fig. 5g and Supplementary Fig. 15), measuring 120 mm × 120 mm x 4.5 mm (length x width x thickness). As a control, we assembled a similar plywood sample with commercial phenol for-maldehyde (PF) resin. The subsequently performed lap shear tests

showed that the plywood bonded with the pulp-IL solution possessed over twofold higher shear strength than the PF bonded plywood, 4.9 MPa as compared to 2.2 MPa. In addition, the pulp-IL bonded ply-wood also showed excellent wet shear strength of 1.7 MPa (Supple-mentary Fig. 16 and Supplementary Table 2). These results further indicate that our wood bonding method is not only applicable for solid wood, but it also has outstanding potential for veneer lamination.

## Discussion

This study demonstrates a multiscale wood bonding method that enables strong interfacial adhesion with excellent water resistance.

This was accomplished by using ionic liquid-dissolved cellulose as the bonding agent during hot-pressing. Among the tested cellulose varieties, pulp with the highest degree of polymerization demonstrated the best performance. Wood bonded with pulp-IL showed shear strength over 20 MPa, exceeding conventional wood bonding methods. Thorough structural and mechanical analyses revealed the critical role of regenerated cellulose and cell wall interlocking in establishing the robust bonding interface. The regenerated cellulose matrix filling the interlocked wood cell walls had a high compatibility with the wood cell wall, facilitating efficient stress transfer that contributed to the superior bonding strength. The outstanding water resistance exhibited by this dense bonding structure effectively addresses the common issue of inadequate water resistance in bio-based adhesives for wood bonding. Moreover, plywood laminated with the pulp-IL solution exhibited nearly three times the bonding strength of conventional plywood bonded with phenol-formaldehyde resin, highlighting the remarkable potential of our bonding method. Thus, our wood bonding method offers a sustainable, high-performance alternative to conventional petroleum-based adhesives used in various engineered wood, mass timber and plywood.

## Methods

### Materials

Sapwood of Scots pine with an oven dry density of 480 kg m$^{-3}$ was obtained from Southeast Finland. Silver birch (*Betula pendula*) veneers with thickness of 1.5 mm and density of 520 kg m$^{-3}$ were provided by Metsä Board, Finland. Microcrystalline cellulose (Avicel PH-101), glycidyl-trimethylammonium chloride, glacial acetic acid, and hydrogen peroxide (35%) were obtained from Sigma Aldrich and used as received. Microfibrillated cellulose and never-dried bleached birch kraft pulp were obtained from Stora Enso, Sweden. 1-Ethyl-3-methylimidazolium acetate, with a minimum purity of 98%, was purchased from Proionic GmbH, Austria. Sodium hydroxide (NaOH pellet) was obtained from VWR.

### Preparation of cationic pulp

The never-dried pulp was first mixed with NaOH to form a suspension with a final concentration of 7.5 wt.% pulp and 5 wt.% NaOH. Glycidyl-trimethylammonium chloride was added to the suspension at a molar ratio of 12.5:1 to the total cellulose anhydrous glucose unit hydroxyl groups, and the reaction was conducted at 65 °C for 6 h with stirring. The reacted suspension was then neutralized with hydrochloric acid, filtered, and thoroughly washed with deionized water. The obtained cationic pulp was freeze-dried and dissolved in [emim][OAc] to form a cationic pulp solution at a concentration of 5 wt.%, following the same procedure as described for the cellulose dissolution section.

### Cellulose dissolution

MFC, pulp, and cationic pulp were freeze-dried for 72 h (− 45 °C and 0.1 mbar) to minimize hornification. Before dissolution, all types of cellulose and [emim][OAc] were subjected to drying in a vacuum oven (VT 6025, Fisher Scientific Oy, Finland) at 70 °C, 200 mbar for 24 h to eliminate any moisture. Subsequently, four different types of 5 wt.% cellulose solution were prepared by dissolving MCC, MFC, pulp, and cationic pulp separately in [emim][OAc] using a vertical kneader system at 85 °C for 2 h. The resulting 5 wt.% cellulose solutions were employed for bonding wood.

### Bonding of wood blocks and plywood production

Cellulose-IL solution was homogeneously applied on the surface of one wood block at a solid spread rate of 9 g m$^{-2}$. Another wood block was placed on top, with direct contact to cellulose-IL solution. The wood assembly was then preheated for 5 min in a hydraulic hot press (Vakomet KRO-260, Lakeuden Hydro Oy, Finland) at 140 °C, followed by compression at 1.5 MPa for an additional 25 min. Subsequently, the hot-pressed wood assembly was rinsed with deionized water to allow the regeneration of cellulose. After the regeneration, the bonded wood assembly was air-dried at ambient conditions (20 °C, 30 % RH), followed by conditioned at 65 % RH and 20 °C until further testing. The resulting samples are denoted as MCC, MFC, and Pulp, respectively. Additionally, a control group was prepared by bonding wood blocks with a 5 wt.% MCC-IL solution without the rinsing step, referred as 'hot-pressing only'. Three-layered plywood was prepared using birch veneer with a 5 wt.% pulp-IL solution, following the same procedure as used for bonding solid wood. Plywood was also prepared by using a commercial PF film and hot-pressed under the same conditions.

### Wiesner staining

Thin transverse sections (thickness: 25 μm) of the wood assembly were prepared using a rotary microtome (ZFP-011, Nahita, Spain). These sections were subjected to the histochemical Wiesner reaction by treating with a 2% phloroglucinol in 95% ethanol solution for 5 min, followed by mounting with a 6 M hydrochloric acid solution. The morphology of the bonding interface was then observed using a light microscope (Olympus, BX53M, Japan).

### Scanning electron microscopy

The microstructure of the bonding interface in wood assembly was observed with SEM (ZEISS Sigma VP, Germany). The sample surface was trimmed and sputtered with a thin layer of Au/Pt (80:20) before conducting SEM imaging.

### Cell wall swelling by cellulose-IL solution

The time- and temperature-dependent swelling of the cell wall with cellulose-IL solution was observed using an Olympus BX53M microscope. A 30 μm-thick transverse section of solid pine wood was covered with a drop of cellulose-IL solution between two glass slides. The incubation temperature was either at 20 °C or 60 °C, and the incubation duration was up to 30–60 min. Subsequently, the wood section was thoroughly rinsed with water.

### Fluorescence microscopy

Fluorescence microscopy was carried out using an Axio Observer Z1 microscope (Carl Zeiss, Germany) equipped with a motorized stage, a 20 × objective, a 1.6 × tube lens, and an Axiocam 503 camera. Typically, a 25 μm-thick transverse section of the bonded wood was stained with calcofluor white. Excess dye was removed by applying a drop of 10% potassium hydroxide. The sample was then observed under blue light ($\lambda_{EX}$ = 420 nm). Image processing and analysis were conducted using Zen Blue software (version 3.5).

### Wide angle X-ray Scattering (WAXS)

The microstructure of the wood adherends' interface was examined by a bench beamline SAXS/WAXS device (Xeuss 3.0 C, Xenocs, France) with Cu Kα radiation (wavelength 1.542 Å). An X-ray beam with an approximate size of 0.4 mm × 0.4 mm was used to measure the scattering at the bonding interface and at a 0.75 mm distance away from it (bulk wood) in radial-longitudinal sections of approximate thickness of 0.7 mm. The samples were in the vacuum of the scattering device. The scattering data were normalized and averaged azimuthally using the XSACT 2.7 software by Xenocs.

### Attenuated Total Reflectance-Fourier transform infrared (ATR-FTIR) spectroscopy

FTIR spectra were obtained using a Spectrum Two FT-IR Spectrometer with a LiTaO$_3$ detector (PerkinElmer, USA). The resolution was 4 cm$^{-1}$, and 32 scans were accumulated over a range of 600–3800 cm$^{-1}$ for each sample.

## Confocal Raman microscopy

Transverse sections (thickness = 25 μm) of wood samples were obtained with a microtome, mounted with deionized water on glass slides, and sealed with nail polish. For Raman mapping, samples were further delignified using a mixture of acetic acid and hydrogen peroxide (1:1 by volume) at 80 °C for an hour, followed by thorough washing with deionized water before mounting on the glass slides. Raman spectra and heat map were obtained with a confocal Raman microscope (Renishaw, inVia™ Qontor, UK) equipped with a 532 nm laser, a 64 × water-immersion objective, and a Centrus 05TJ55 CCD detector behind a 2400 lines mm$^{-1}$ grating. Raman mapping was performed with 349 lines per image and 226 points per line at a step size of 0.4 μm. The integration time for Raman imaging was set to 0.5 s at 100% laser intensity.

The Raman images were fused into a mosaic and reshaped from the 3D structure into a 2D array, with each pixel represented as a row and the spectral information in the columns. Two separate wavenumber regions were extracted, corresponding to 750–780 cm$^{-1}$ and 1550–1700 cm$^{-1}$. A linear baseline correction ($n = 1$) was applied to the extracted datasets separately, and the area under the curve was calculated using the trapezoidal method. This method numerically approximates the area by dividing the region into trapezoids, providing a robust metric to compare the overall signal intensities. The calculated area values were reshaped back into 2D image dimensions, and the scale was adjusted to visualize the differences.

Cell corners and middle lamella spectra were extracted by carefully selecting regions of interest (ROIs) from the 3D Raman image data. The spectral data from multiple ROIs (including pixel indices and corresponding spectra) were arranged into a 2D array. Wavenumbers outside the 540–1725 cm$^{-1}$ range were excluded. Baseline correction was performed using a 5th-degree polynomial subtraction, followed by smoothing using a Savitzky-Golay[37] procedure (polynomial order $n = 2$, 15-point window). The spectra were then normalized using a unit vector, and the mean spectrum was extracted for comparison. All spectral processing was performed using in-house MATLAB 2023a (MathWorks, Inc.) scripts.

## Viscosity-average degree of polymerization

The DP$_v$ of the respective cellulose was determined from the intrinsic viscosity number measured according to SCAN-CM 15:88 standard.

## Viscosity of cellulose-IL solutions

The viscosity of cellulose-IL solution was determined using an Anton Paar MCR302 stress-controlled rheometer equipped with plate-plate geometry (25 mm in diameter) and a Peltier temperature control setup. Temperature-dependent viscosity measurements were conducted at 20 °C and 140 °C using a logarithmic shear rate ramp from 0.01 to 10 s$^{-1}$. A total of 23 data points were recorded within this shear rate range, with a dwell time of 10 s per point.

## Tensile shear test of bonded wood assembly

The shear strength of the bonded wood assembly was evaluated in accordance with the type A1 (A5 for delamination test of Pulp-bonded sample) of EN 302-1:2023, with slight modification. The test specimens had a dimension of 150 × 15 × 20 mm³ (length × width × thickness) (Supplementary Fig. S8). The test was done using a universal material testing device (Zwick 1475, Germany) at an operating speed of 5 mm min$^{-1}$. The shear strength of the wood bonding line was determined in the dry state after one-week conditioning in a climate room (20 °C, 65% RH). The shear strength was calculated using Eq. 1.

$$F = F_{max} A^{-1} \tag{1}$$

Where $F$ denotes the shear strength in MPa, $F_{max}$ represents the maximum applied load (N) at failure, and A is defined as the bonded test surface area in square millimeters. The test specimen is illustrated in Fig. 5a.

For the assessment of shear moduli, the specimens were recorded with an EOS R8 frame (Canon, Japan) equipped with an RF 24–105 mm F4L IS USM lens (Canon, Japan) during the shear tests. Deformation of the specimens in the bonding region was determined by digitally tracking the movement of two pairs of points (see Fig. 5a and Supplementary Movie 2), each pair separated by the bonding line. The recorded videos were processed with an in-house image point-tracking script, yielding coordinates of the tracking points for each recorded frame[38]. This data was processed and used to calculate the shear stiffness according to Eq. 2.

$$G = F \, l \, (A \, \Delta x)^{-1} \tag{2}$$

Where $G$ denotes the shear stiffness in MPa, $\Delta x$ is the vertical displacement of two opposite points, and $l$ is the initial horizontal distance between these same points.

Systematic rotation of all four points due to the natural deformation of the specimens subjected to increasing shear stress was considered by correcting vertical displacement for the rotation angle θ, defined from averaging the angles of the virtual, vertical lines of points, each on the same side of the bonding line. The resulting strain data was subsequently combined with stress data from the material testing device, the slope of the resulting stress-strain curve within the linear elastic region (0-1% strain) gave the shear stiffness.

## Data availability

The data that support the findings of this study are available from the corresponding author upon request, and the source data are present. Source data are provided in this paper.

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

## Acknowledgements

This work made use of Aalto University's OtaNano-Nanomicroscopy Center Facilities. This work was partially funded by Business Finland (6910/31/2023) and the Research Council of Finland (341701). S.Z. is grateful to the financial support from the Finnish Foundation for Tech-nology Promotion (10769), Aalto-yliopiston tekniikan tukisäätiö (20250009), Puunjalostus Insinöörit, Eero Kivimaan Stipendir-ahastosäätiö, and Puumiesten Ammattikasvatus Säätiö (2025). P.P. acknowledges the Research Council of Finland for funding (338804). M.A. acknowledges the funding from the Finnish Cultural Foundation (00240121) and Tutkijat Maailmalle (20240032). The authors gratefully acknowledge Joonas Jaaranen for granting permission to use the digital image point-tracking algorithm.

## Author contributions

L.R., S.W., and S.Z. conceptualized the project. L.R. and S.W. supervised the project. S.Z. prepared all samples and performed light microscopy, Raman microscopy, FTIR, SEM, rheology measurement, and tensile shear test. S.K. performed fluorescence microscopy and conceptualized the cellulose and ionic liquid behavior during hot pressing. H.M. con-tributed to the tensile shear test using digital image correlation. P.P. performed WAXS analysis. M.A. contributed to the processing and interpretation of Raman microscopy data. The manuscript was primarily written by S.Z., S.W., and S.K. in consultation with L.R. and M.L. All authors have read and agreed to the submitted version of the manuscript.

## Competing interests

The authors declare no competing interests.

## Additional information

**Peer review information** *Nature Communications* thanks Yi Hong, Shaoliang Xiao, and the other anonymous reviewer(s) for their con-tribution to the peer review of this work. A peer review file is available.

