## [Transparent Peer Review file · Nature Communications]

Multiscale interface engineering enables strong and water-resistant wood bonding

Corresponding Author: Dr Shennan Wang

Version 0:

Reviewer comments:

Reviewer #1

(Remarks to the Author)

[Note from the editor: see comments in the attached document.]

Reviewer #2

(Remarks to the Author)

This article reports on the use of cellulose-ionic liquid solutions to achieve high-performance bonding in wood. The experimental design is well-organized, and different characterization methods are employed to illustrate the bonding mechanism of cellulose-ionic liquid solutions in wood. However, several issues require revision and clarification.

1、 The approach of preparing wood adhesives through the dissolution and regeneration of cellulose using ionic liquids or NaOH/urea/H₂O have already been reported. The authors need to further clarify the novelty of their work.

Wang, S., Chen, M., Hu, Y., Yi, Z., & Lu, A. (2024). Aqueous Cellulose Solution Adhesive. *Nano Letters*, 24(19), 5870-5878. doi:10.1021/acs.nanolett.4c01154

Huang, Z., Cao, Z., Huang, D., Li, S., Zhu, M. and Chen, Y. (2024), Bonding Wood via Cellulose Aqueous Solution as Cell Wall Adhesive. *Adv. Eng. Mater.*, 26: 2301719. <https://doi.org/10.1002/adem.202301719>

2. On page 2, lines 16-22, the authors state that the bonding of the adhesive to the wood is achieved by the adhesive flowing into the multiscale pores of the wood, which is correct. However, their further assertion that wood bonding with adhesives cannot enable the bonded wood to surpass the mechanical performance of solid wood is incorrect. Theoretically, when the failure mode of a lap joint is substrate failure, the strength of the bonded wood can surpass that of solid wood. Furthermore, cases where adhesives have widely enabled bonded wood to surpass the mechanical properties of solid wood have been reported in other studies.

3、 On page 4, line 5, although the article demonstrates that the Cellulose-IL solution exhibits excellent bonding performance, the authors' claim that this method surpasses all existing bonding methods is overly absolute. At least numerically, there are adhesives with superior performance compared to the Cellulose-IL solution, such as epoxy resin.

4、 The authors used a hot pressing temperature of 140°C. However, does the bonding performance of the Cellulose-IL solution change with higher or lower hot-pressing temperatures?

5、 Besides increasing the flowability of the cellulose solution, are there any other potential chemical changes or crosslinking reactions during hot pressing?

6、 It is clear that water regeneration of cellulose plays an important role in improving the bonding performance of wood.

However, how long does this process take? Additionally, since wood needs to be used in a dried state, does the treated wood require drying afterward? These points may require further clarification.

7、 From the image in Figure 4f, the high-viscosity pulp shows better permeability than the low-viscosity MFC and MCC, as it fills more wood cells lumens. However, this seems counterintuitive because, theoretically, high-viscosity adhesives should have poorer permeability, which would not be conducive to filling more wood cell lumens. This may require further explanation from the authors.

8、 The authors prepared three-layer plywood to evaluate the application potential of pulp-IL. However, in addition to the dry strength tests, the wet strength of the plywood should also be included to provide a more comprehensive evaluation of the bonding performance of pulp-IL.

9、 Ionic liquids are usually expensive, so their recyclability is crucial. After bonding the wood, can the ionic liquid in the cellulose solution be recovered and reused?

10、 1. The viscosity, dissolution capability, and stability of different ionic liquids may significantly impact bonding performance. This may require further elaboration on the criteria used for selecting the ionic liquids.

11、 The numbering of figures in the manuscript contains errors and should be corrected. For example, on page 13, line 23, the figure reference should be corrected to "Fig. S9."

12、 In Figure 5, the material names described in the main text are not consistent with those shown in the figure legend and need to be corrected.

Reviewer #3

(Remarks to the Author)

Zhang and coworkers have submitted a very interesting and well-elaborated manuscript on a new concept to achieve strong wood bonding without using petroleum-based adhesives. The topic is timely and highly relevant because petroleum-based adhesives strongly contribute to the carbon footprint of wood-based products and are one of the decisive factors for the rather low recycling rate of wood. The achieved properties are extremely impressive, in particular the wet strength properties, which are usually the weak spot of alternative wood gluing systems. The comprehensive characterization provides important information on structural and chemical features as well as the resulting mechanical properties. However, these findings can only partly explain the obtained mechanical performance. Specific comments:

Page 2, line 4: Better "carbon storage" than "carbon sink", since sink refers to a different time frame.

Page 2, lines 10-11: I think the argument of limited bond strength of petroleum-based adhesives is questionable since several of these products are established in the timber engineering industry.

Page 3, 9ff: I am wondering whether friction welding is an appropriate reference system for this approach.

Page 4, line 15: the authors claim scalability and implementation potential for the wood industry. Therefore, they should discuss the press conditions in terms of temperature, pressure, and duration. Related to implementation, the relevance of less perfect surfaces should be addressed, in particular the influence of knots.

Page 5, lines 2-4: "Notably, the cell wall of both adherends intertwined, forming a mechanically interlocking structure resembling that of friction-welded wood." I am skeptical about this statement since the loading cases are entirely different. The authors may better explain the nature of the intertwined structures.

Page 5, lines 17-18: it should be added that staining does not allow for a quantification.

Page 6, lines 20-21: I would expect earlywood cell walls to be thinner. Did you measure individual cell walls or double cell walls (lumen to lumen)?

Page 7, lines 5-11: I am very surprised that the authors did not detect cellulose II. The authors should conduct an XRD analysis with a sample cut into small pieces to eliminate the structural orientation pattern.

Page 9, lines 14-16: A penetration through pits is known for other adhesives as well. A peculiar feature of many formaldehyde-based adhesives is their ability to penetrate the cell wall.

Page 13, lines 2-6: The used pine species specimens possess large, window-like, cross-field pits, which may also play an important role in the transverse penetration processes. This should be tested by doing a control experiment with a species like spruce.

Page 13, lines 17-19: I find this statement on the "interconnection" ability a bit vague. What is the actual mechanism? What is the type of interaction between the cell wall and the added cellulose?

Page 14, lines 9-13 and 16-18: The achieved shear strength is very remarkable, but it appears questionable, how it can be that high. If a failure appears in the bulk (and with a certain distance from the bondline), the shear strength of the wood is the determining factor. In the test setup, the shear strength of pine wood should be around 10 MPa. Even if one considers a certain level of densification these values are still far off. The authors need to provide a more elaborated interpretation of how such high values can be obtained, which goes beyond the rather vague "multiscale mechanism" addressed on Page 15, lines 7-12.

Page 16, lines 1-6: The obtained wet shear strength is extremely impressive. Again, I see the problem that these values are far above the shear strength of wet wood. Additionally, the in-depth characterization of the bondline does not provide

sufficient evidence for such extraordinary water resistance. Do we miss an important piece of the puzzle? Tensile-shear tests are a good performance indicator but for the claimed implementation potential delamination tests in drying and wetting cycles are by far more relevant. These tests should be conducted.

Version 1:

Reviewer comments:

Reviewer #1

(Remarks to the Author)

The authors have provided a comprehensive and well-structured response to the concerns raised in the previous review. In particular, the additional data and analyses presented on the mechanical performance and adhesion mechanisms significantly strengthen the scientific rigor and clarity of the manuscript. The newly added mechanical characterization is both quantitative and comparative, offering a more robust validation of the material properties. Furthermore, the improved discussion and supporting evidence regarding the adhesion mechanisms enhance the mechanistic understanding and provide greater persuasive power to the overall conclusions. At this stage, I have no further concerns. I believe the manuscript has been substantially improved and now meets the standards for publication in Nature Communications. I recommend acceptance.

Reviewer #2

(Remarks to the Author)

The authors have satisfactorily addressed all the points I raised during the review. I recommend the manuscript for publication in its current form.

Reviewer #3

(Remarks to the Author)

The authors conducted additional experiments and have convincingly responded to the raised concerns. No further comments.

Response to the comments

We thank the reviewers for their valuable comments and constructive feedback, which have helped us improve the manuscript. We have made every effort to address all the points raised and have revised the manuscript accordingly. Please find below a point-by-point response to the reviewers' comments. The reviewers' comments are highlighted in **blue**. Our responses are in **black** and the corresponding revisions in the manuscript are highlighted in **red**.

Reviewer #1 (Remarks to the Author):

The manuscript presents an innovative strategy to bond wood using a solution of cellulose in an ionic liquid. This approach is interesting and has potential implications for sustainable material development. However, there are significant issues with the experimental design of the mechanical performance tests, which compromise the validity of the results regarding the adhesive's bonding strength. Without addressing these critical issues, the discussion of the bonding strength of the adhesive remains unsubstantiated. I recommend a major revision. Only after resolving the following issues can the manuscript be reconsidered for publication.

Response: We thank the reviewers for recognizing the novelty and potential of our work. We also appreciate the constructive critique regarding the experimental design. In response, we have carefully revisited the mechanical performance testing methodology to ensure that the reported results and associated discussion are based on a reliable experimental design.

1. In all tensile shear tests conducted, failure occurred in the bulk wood rather than at the bonding interface. This indicates that the experimental setup does not accurately reflect the adhesive's shear strength. Specifically, the maximum force corresponding to reported approximately 23 MPa shear stress still reflects fracture force of the bulk wood, which is approximately twice the tensile strength of wood. This outcome is expected based on the test specimen design illustrated in Fig. S9, where the tensile area of the wood is doubled while the bonded area matches the cross-sectional area of a single wood specimen. While the data demonstrate that the adhesive likely exhibits a shear strength exceeding the intrinsic tensile strength of wood, the absolute values in Fig. 5b are not comparable across different experimental groups. Consequently, the performance of various types of cellulose in influencing the adhesive's properties cannot be reliably assessed. Furthermore, any interpretation of Fig. 5, as well as its correlation with Fig. 4, must be revisited once the accurate

shear strength is measured. I am unsure whether the authors referenced any ISO or ASTM standards for their testing. However, the experimental design could be modified to better align with the adhesive's material properties. For example, reducing the bonded area would ensure that the bonding interface fails under lower tensile forces, avoiding bulk wood fracture and yielding data more representative of the adhesive's true performance.

Response: For the determination of longitudinal tensile shear strength of bonded wood, we followed the European standard EN-302-1:2023, which is also referenced in the international standard ISO 20152-1:2010. Deviating from the standard, the thickness of bonded specimen was increased from 5 to 10 mm. The tensile shear strength of wood was determined using the same sample size as the bonded wood.

We agree that sample geometry, particularly overlap length, can influence the test result. In response to Reviewer #1's suggestion, we prepared specimens with both the 10 mm standard overlap length and a 6 mm reduced overlap length (**Fig. R1**), thereby varying the bonding area for the tensile shear tests. For positive control group (wood thermally treated at 140 °C for 30 min), a tensile shear strength of 13.9 ± 0.5 MPa was obtained when 10 mm overlap length was used. However, when the overlap length was reduced to 6 mm, the tested tensile shear strength was recorded at 20.4 ± 1.5 MPa, with a significant increase by 47% compared to those with a standard 10 mm overlap length. Similar phenomenon was also observed for bonded wood. For wood bonded with 5 wt.% pulp-IL solution, the tensile shear strengths of 5 replicates were 20.1 ± 0.7 MPa and 28.8 ± 3.2 MPa when the overlap lengths were 10 mm and 6 mm, respectively. This outcome is consistent with findings reported earlier by W. Gindl-Altmutter, U. Müller, J., and Konnerth, in which the measured shear strength of polyvinyl acetate bonded beech wood decreased significantly (from 10 to 5 MPa) with increasing overlap length (5, 10, 24 mm), due to the increased stress concentration factor by two orders of magnitude. Although the result obtained with 6 mm overlap length may reflect better the performance of our bonding method possibly because of a more even stress distribution during test, the use of a geometry that differs from the majority of literature also limits the comparability of our results. Therefore, we continued to use the standard 10 mm overlap length for all subsequent samples when re-evaluating wood bonding strength.

Reference: W. Gindl-Altmutter, U. Müller, J., and Konnerth. The significance of lap-shear testing of wood adhesive bonds by means of Volkersen's shear lag model. *European Journal of Wood and Wood Products* 70, 903-905 (2012). doi: 10.1007/s00107-012-0621-z.

Fig. R1. Influence of overlap length on the measured shear strength of the positive control (thermally treated wood) and wood bonded with 5 wt.% pulp-IL solution. Yellow filled circles represent shear strength measured from samples with a 10 mm overlap length along the bonding line, whereas green filled circles correspond to shear strength measured from samples with a 6 mm overlap length along the bonding line.

In our original experimental design, a longer sample length was used for Pulp-IL bonded wood to enhance gripping and to ensure appropriate failure during tensile shear test. In response to Reviewer #1's concern on the different sample geometries used across sample groups, we prepared a new batch of bonded wood having the same nominal size of 150 mm × 20 mm × 15 mm (length × thickness × width). The newly measured shear strength and shear stiffness have been updated to Supplementary Table 2. The new stress-strain curves obtained by point-tracking have been updated to Figure 5b. As shown below and also in Supplementary Fig. 10, the shear strength of solid wood remained consistent with our last results at ca. 12.2 MPa. The newly measured shear strength of MCC, MFC, and Pulp bonded wood samples decreased slightly as compared to the previous results, while maintaining the same trend. This is attributed to the increased overlap length (from 8 to 10 mm) and the improved groove cutting as suggested in an earlier study to ensure reliability of the results.

Reference: Ramachandrareddy, B., Solt-Rindler, P., Herwijnen, H. WG. van, Pramreiter, M. & Konnerth, J. Sensitivity of lap-shear test to errors in groove cutting and influence of wood type/treatment. *Int. J. Adhes. Adhes.* 130, 103605 (2024).

Supplementary Fig. 10. Shear strength of wood bonded with different cellulose-IL solutions and corresponding photos of fractured samples. Horizontal lines indicate mean value.

2. The manuscript inconsistently refers to the adhesive as "cellulose-IL" and "pulp-IL," for example, in lines 18 and 20 on page 6. Please standardize terminology or provide clear explanation.

Response: In the original manuscript, 'cellulose-IL' referred to pulp cellulose-ionic liquid solution. To differentiate between the three types of cellulose sources used, each adhesive was later named according to its specific cellulose type (e.g., pulp-IL or MCC-IL). To avoid inconsistency, we have now revised all instances of 'cellulose-IL' to reflect the exact cellulose source throughout the manuscript.

3. In **Supplementary Table 1** and **Fig. S3**, the description of earlywood cell wall swelling in ionic liquid is based on measurements of five locations on the cell wall from the same sample. However, for the results to hold statistical significance and represent general trends, the selected cell wall measurements should ideally be derived from multiple samples and multiple cell walls. Relying on a single sample limits the robustness of the data and may introduce sampling bias.

Response: In addition to the original sample, we conducted measurement on three additional wood sections. For each section, measurements were taken at five random spots in both earlywood and latewood regions. The mean values and standard deviations of the compound cell wall thickness are reported in **Supplementary Table 1**. **Supplementary Fig. 3** is retained as an example of the observed cell wall swelling and reformation behavior.

4. The manuscript employs two distinct approaches to investigate the distribution of cellulose within wood: cationic modification of cellulose in **Fig. 3** and characteristic staining of cellulose in **Fig. 4**. However, the rationale for using different methods in these experiments is unclear. The authors should clarify why a staining-based method was not employed in **Fig. 3**, as it appears to be a more direct and less disruptive approach for visualizing cellulose distribution. The use of cationic-modified cellulose could interfere with cellulose-lignin interactions, potentially altering the natural distribution of cellulose. Additionally, the delignification process may have affected the cellulose structure, introducing further uncertainties. These factors need to be carefully considered and discussed to validate the results presented in **Fig. 3**.

Response: We understand Reviewer #1's concern regarding the use of different microscopy methods in this study. Raman microscopy, with its ability to provide detailed chemical information, is particularly effective for resolving the distribution of modified cellulose bearing specific functional groups that differ from those in native cellulose within the cell wall. This

made it suitable for investigating the penetration behavior of cation-modified pulp and the interactions between the adhesive and wood during the bonding process. In contrast, fluorescence microscopy was employed to rapidly visualize the distinct behaviors of various cellulose-IL adhesives in filling the cell lumina without the need to chemically modify cellulose. Given its limitation in distinguishing native cellulose from regenerated cellulose, calcofluor white staining is more appropriate for identifying lumen-filling structures rather than resolving detailed features within the cell wall. Nevertheless, both methods are complementary, offering insights at different spatial and chemical resolutions.

We acknowledge that cation-modified cellulose could potentially interfere with the cellulose-lignin interactions and alter the natural distribution of cellulose. However, the differentiation between native cellulose and added cellulose is very challenging, especially when the regeneration process didn't lead to a high conversion rate to cellulose II. The introduction of cation as a labelling tag helped with the elucidation of cellulose penetration pathway, which is important in explaining the high shear strength we have obtained in this study. We have added potential difference between cationic pulp and non-modified pulp in distribution and cellulose-lignin interaction as a fact in the manuscript. The manuscript is revised as follow on Page 8, line 12-14:

... Although cationic modification of cellulose may alter its distribution and interactions with lignin, this approach provides valuable spatial and chemical insights into the formation of bonding interface

To assess the influence of delignification process on the structure of cellulose, we further subjected cationic pulp-IL solution first to water-induced regeneration and then delignification with H₂O₂/acetic acid. FTIR result (**Fig. R2**) showed that the intensity of characteristic peak of trimethylammonium chloride at 1478 cm⁻¹ remains unchanged. And the characteristic band of -OH in 3500-3000 cm⁻¹ resembles cellulose II that found in regenerated pulp (Supplementary Fig. 4). The new peak showing up at 1570 cm⁻¹ is possibly due to the ionic complexation between trimethylammonium cation and acetate anions from ionic liquid. These suggest that the delignification would not affect the detection of cationic cellulose.

Fig. R2. Attenuated Total reflectance-Fourier Transform Infrared (ATR-FTIR) spectra of pulp, cationic pulp, and the cationic pulp after regeneration and bleaching with hydrogen peroxide (H_2O_2) and acetic acid mixture (1:1 v/v) for 1 hour at 80 °C.

5. I recommend a deeper investigation into the interaction mechanisms between cellulose in ionic liquid (IL) and wood. Intuitively, this interaction depends on the diffusion of cellulose during the hot-pressing process and the phase separation induced by water. While the authors have explored the influence of molecular weight on cellulose distribution, primarily in the context of the diffusion process, the impact of this factor on mechanical performance requires reexamination and validation. Beyond molecular weight, it would be favorable to investigate additional factors that may significantly contribute to the adhesive performance. What I can think of includes the concentration of cellulose in IL and the kinetics of regeneration induced by water, which might influence the bonding behavior. These parameters could play critical roles in determining the distribution, interaction, and eventual performance of the adhesive. If the authors could systematically elucidate how such factors affect the cellulose-wood interaction and establish a correlation with the mechanical results, they would provide a more comprehensive and robust mechanism for this bonding strategy. Such insights would greatly enhance the scientific significance of the study.

Response: In addition to molecular weight, we further investigated the effects of pulp concentration and regeneration (phase separation) methods on wood bonding with pulp-IL

solutions. The results of tensile shear test, along with corresponding photos of fractured specimens, have been included in Supplementary Fig. 17-20.

We first examined the flow behavior of pulp-IL solution at difference pulp concentrations (Supplementary Fig. 17). At 20 °C, the steady state viscosities of 3, 5, and 8 wt.% pulp-IL solutions differed by approximately one order of magnitude. However, at 140 °C, the viscosities dropped significantly, and the difference between the 5 and 8 wt.% solutions became negligible. The difference in viscosity is expected to influence the penetration of pulp-IL solutions into wood. As shown in the fluorescence microscopy images (Supplementary Fig. 18), the wood bonded with 3 wt.% pulp-IL solution had a thicker bonding interface with regenerated cellulose primarily detectable at the lumina surface. This suggests better penetrability into the wood structure due to the much lower viscosity but a reduced capacity for network formation due to the lower cellulose concentration. In contrast, wood bonded with 5 wt.% and 8 wt.% pulp-IL solutions showed thinner bonding interface and fully filled lumina. The lumen filling behavior of 8 wt.% pulp-IL closely resembled that of 5 wt.% pulp-IL.

Due to differences in lumen filling, wood bonded with 3 wt.% pulp-IL showed lower shear strength of 15.9 ± 1.1 MPa, while 8 wt.% pulp-IL bonded wood reached 20.3 ± 0.5 MPa, slightly higher than the result obtained with 5 wt.% pulp-IL (Supplementary Fig. 19). In conclusion, pulp concentration lower than 5 wt.% may impair wood bonding performance due to limited network formation in the lumina at the bonding interface. A pulp concentration higher than or equal to 5 wt.% is required to achieve the best bonding performance.

To investigate the kinetics of cellulose regeneration induced by water, we performed regeneration process by using two alternative methods: 1) rapid regeneration by exposure to water steam (50 mL per sample) with an iron, and 2) slow regeneration in a 65% RH environment for one month (Supplementary Fig. 20). Both methods resulted in lower shear strength (11.6 ± 1.2 MPa and 13.2 ± 0.7 MPa, respectively) compared to regeneration with water rinsing for overnight (19.6 ± 1.3 MPa). However, the bonding strengths achieved with both methods were still comparable to that of solid wood. These findings indicate that the effectiveness of the regeneration process significantly influences the overall bonding performance.

Supplementary Fig. 17. Flow curves of pulp-IL solution containing 3, 5, and 8 wt.% pulp at 20 °C and 140 °C.

Supplementary Fig. 18. Fluorescence microscopy images showing calcofluor white stained cross-sections of wood bonded using 3, 5, and 8 wt.% pulp-IL solutions.

Supplementary Fig. 19. The summary of shear strength and the photos of fractured wood samples bonded with different concentrations of pulp-IL solution and hot-pressed for 30 min at 140 °C with 1.5 MPa pressure. a) wood sample bonded with 3 wt.% pulp-IL solution. b) wood sample bonded with 8 wt.% pulp-IL solution.

Supplementary Fig. 20. The bonding strengths and the fracture images of wood samples bonded with 5 wt.% pulp-IL solution hot-pressed for 30 min at 140 °C with 1.5 MPa pressure. a) bonded wood regenerated with steam. b) Bonded wood sample regenerated in 65 % RH

Reviewer #2 (Remarks to the Author):

This article reports on the use of cellulose-ionic liquid solutions to achieve high-performance bonding in wood. The experimental design is well-organized, and different characterization methods are employed to illustrate the bonding mechanism of cellulose-ionic liquid solutions in wood. However, several issues require revision and clarification.

Response: We thank the reviewer for recognizing our work. We have made all possible efforts to address the issues raised and to improve the clarity of the manuscript.

1. The approach of preparing wood adhesives through the dissolution and regeneration of cellulose using ionic liquids or NaOH/urea/H₂O have already been reported. The authors need to further clarify the novelty of their work.

Wang, S., Chen, M., Hu, Y., Yi, Z., & Lu, A. (2024). Aqueous Cellulose Solution Adhesive. *Nano Letters*, 24(19), 5870-5878. doi:10.1021/acs.nanolett.4c01154

Huang, Z., Cao, Z., Huang, D., Li, S., Zhu, M. and Chen, Y. (2024), Bonding Wood via Cellulose Aqueous Solution as Cell Wall Adhesive. *Adv. Eng. Mater.*, 26: 2301719. <https://doi.org/10.1002/adem.202301719>

Response: We appreciate the reviewer for highlighting these two relevant references. We were aware of both studies and have carefully considered the distinction between their approaches and ours. The synthesis of cellulose solutions and their application in adhesion is not a new concept. In literature, numerous studies have reported the use of cellulose-based adhesives, including those derived from cellulose derivatives and nanocellulose. For example, cellulose nitrate has long been used as a general-purpose adhesive, involving its dissolution in a solvent and subsequent regeneration through solvent evaporation. The works by Wang *et al.* and Huang *et al.* followed a similar concept, using different cellulose solvents—BzMe₃NOH/H₂O and NaOH/urea/H₂O, respectively. However, the primary focus of our study is not the development of a new adhesive, but rather the introduction of a novel wood bonding method. Our approach results in a unique bonding interface structure, characterized by interlocked wood cell walls and lumina filled with a regenerated cellulose matrix. This interface structure, which plays the critical role in providing high wood bonding strength, was not observed in the aforementioned studies.

To acknowledge the existing works in bonding wood with ionic liquid and cellulose solutions and to clarify the novelty of our own contributions, we have included the following discussion in the revised manuscript (Page 16, lines 10-14):

...Although there have been attempts to use either ionic liquids (Nakaya *et al.*) or cellulose solutions (Wang *et al.* and Huang *et al.*) for wood bonding, none of these studies successfully achieved the formation of an interlocked cell wall structure combined with a regenerated polymer network filling the cell lumina—features that are critical to the high bonding performance demonstrated in this work. ...

2. On page 2, lines 16-22, the authors state that the bonding of the adhesive to the wood is achieved by the adhesive flowing into the multiscale pores of the wood, which is correct. However, their further assertion that wood bonding with adhesives cannot enable the bonded wood to surpass the mechanical performance of solid wood is incorrect. Theoretically, when the failure mode of a lap joint is substrate failure, the strength of the bonded wood can surpass that of solid wood. Furthermore, cases where adhesives have widely enabled bonded wood to surpass the mechanical properties of solid wood have been reported in other studies.

Response: The reviewer is correct that in some cases the petroleum-based adhesives could result in mechanical strength exceeding the wood substrates. In our original statement, the mechanical performance is referred to the shear strength of the bonding line, therefore, we have changed the term of ‘*mechanical performance*’ to ‘**shear strength**’ on page 2, line 22. Furthermore, since there exist cases that bonding strength is slightly higher than the wood substrates, we changed the ‘neither method enables’ into ‘neither method **ensures**’ the bonded wood to surpass the shear strength of solid wood on page 2, lines 21.

3. On page 4, line 5, although the article demonstrates that the Cellulose-IL solution exhibits excellent bonding performance, the authors' claim that this method surpasses all existing bonding methods is overly absolute. At least numerically, there are adhesives with superior performance compared to the Cellulose-IL solution, such as epoxy resin.

Response: To avoid overstatement, we have revised the original claim to ‘outperforming **many of the** existing bonding methods’ on Page 4, line 5.

4. The authors used a hot-pressing temperature of 140 °C. However, does the bonding performance of the Cellulose-IL solution change with higher or lower hot-pressing temperatures?

Response: We bonded wood using 5 wt.% pulp-IL solution at both room temperature (20 °C) and 90 °C, right above the dissolution temperature for cellulose (85 °C) (Supplementary Fig. 21), both at a pressure of 1.5 MPa. Notably, the wood samples bonded at 20 °C exhibited delamination during water regeneration, indicating insufficient interfacial adhesion at low temperature. The wood samples bonded at 90 °C demonstrated a reduced bonding strength of 11.5 MPa compared to 19.6 ± 1.3 MPa of those bonded at 140 °C and 1.5 MPa. These results suggest that effective wood bonding requires sufficient heat to promote cellulose penetration and to activate the softening effect of the IL on cell wall components at the bonding interface.

We did not conduct wood bonding at temperature above 140 °C. This is because 140 °C is already considered a relatively high temperature in wood bonding applications. Moreover, it exceeds the depolymerization temperature of hemicellulose, beyond which significant wood degradation may occur. Therefore, we believe it is more relevant to explore the effects of lower bonding temperatures which are more applicable to practical and industrial settings.

Supplementary Fig. 21. The shear strength and photos of fractured wood samples bonded with 5 wt.% pulp-IL solution and hot-pressed at different temperatures for 30 min. a) wood sample hot-pressed at 90 °C with 1.5 MPa pressure. b) wood sample hot-pressed at 20 °C with 1.5 MPa pressure.

5. Besides increasing the flowability of the cellulose solution, are there any other potential chemical changes or crosslinking reactions during hot pressing?

Response: We agree that there are potential chemical changes during hot pressing. But the enhancement is expected to be unsubstantial. We conducted tensile shear strength test on

thermally treated wood (140 °C, 30 min). The results show a slight increase in shear strength for thermally treated control (13.9 MPa) compared to the untreated control (12.2 MPa) (Fig. R3). This observation aligns with the findings by Al-musawi, H., Manni, E., Stadlmann, A. *et al.* Characterisation of thermally treated beech and birch by means of quasi-static tests and ultrasonic waves. *Sci. Rep.* **13**, 6348 (2023). doi: 10.1038/s41598-023-33054-w, where thermally treated beech and birch exhibited moderate improvements in shear strength and stiffness following treatment at temperature below 200 °C. The enhancement was attributed to condensation reaction of lignin.

Furthermore, FTIR spectra were collected for both untreated control and the thermally treated control solid wood. A slight reduction in the hydroxyl peak around 3350 cm⁻¹ was observed in the thermally treated wood, suggesting that additional chemical changes or crosslinking reactions may have occurred during hot pressing. These changes likely contribute to the modest increase in wood shear strength.

Fig. R3. The shear strength and photos of fractured wood samples and the corresponding FTIR spectra.

6. It is clear that water regeneration of cellulose plays an important role in improving the bonding performance of wood. However, how long does this process take? Additionally, since

wood needs to be used in a dried state, does the treated wood require drying afterward? These points may require further clarification.

Response: For our standard samples (140 °C, 30 min, 1.5 MPa, and shear strength of 19.6 ± 1.3 MPa), the water regeneration was performed overnight. We have also tested two additional methods to induce regeneration: by water steam to quickly induce regeneration and by slow regeneration in 65% RH environment for one month. Both methods were tested with wood bonded at 140°C, 1.5 MPa, 30 min, and their shear strength (Supplementary Fig. 23) were lower than the ones regenerated with water rinsing.

After regeneration process, the bonded wood was air-dried under ambient conditions (20 °C and 30% RH). This clarification has been added to the Bonding of wood blocks and plywood production section in the Materials and Methods (Supplementary Information).

7. From the image in Figure 4f, the high-viscosity pulp shows better permeability than the low-viscosity MFC and MCC, as it fills more wood cells lumens. However, this seems counterintuitive because, theoretically, high-viscosity adhesives should have poorer permeability, which would not be conducive to filling more wood cell lumens. This may require further explanation from the authors.

Response: We agree that high-viscosity pulp should exhibit poorer permeability. We measured the thickness of lumina filled regions in bonded wood (**Fig. R4**) and found out a thicker filled region for MFC ($\approx 130 \mu\text{m}$) than pulp ($\approx 110 \mu\text{m}$), indicating better permeability of MFC-IL solution. However, no lumina filled with regenerated MCC were observed, making it difficult to observe the penetration distance of low molecular weight MCC. As we have stated in the original manuscript, low molecular weight cellulose, while capable of deeper penetration into the wood structure, was likely washed away during the rinsing with water. Its low molecular weight also impairs the network formation ability due to their shorter polymer chains.

To validate our hypothesis that the lower molecular weight cellulose (i.e. MCC) may have been washed away during the regeneration process, we bonded wood using a 5 wt.% MCC-IL solution under lower pressure (1 MPa) to minimize collapsing of lumina. Instead of direct regeneration in water, the bonded wood sample was conditioned in a 65 % RH environment for a month to promote gradual regeneration and network formation while reducing cellulose loss during subsequent washing. A thin slice of bonded wood was prepared using a microtome and then immersed in water to remove ionic liquid. The morphology of the bonding interface

was examined using calcofluor white staining and fluorescence microscopy. As shown in the fluorescence images (**Fig. R5**, right), most cell lumina at the bonding interface showed regenerated cellulose located at the lumen surface, proving that MCC can travel through several cells and penetrate into wood structure. The thickness of the region with regenerated cellulose reached 190 μm . However, MCC was not able to form gel network to fully fill the lumina limited by its significantly lower molecular weight, which consequently led to reduced shear strength of the MCC-IL bonded wood.

Fig. R4. Fluorescence microscopy images showing calcofluor white stained cross-sections of wood bonded using various celluloses.

Fig. R5. Fluorescence microscopy images showing the calcofluor white stained cross-sections of wood bonded with a 5 wt.% MCC-IL solution.

8. The authors prepared three-layer plywood to evaluate the application potential of pulp-IL. However, in addition to the dry strength tests, the wet strength of the plywood should also be included to provide a more comprehensive evaluation of the bonding performance of pulp-IL.

Response: The wet bonding strength of the plywood was measured, yielding a mean value of 1.7 MPa. This result is included in Supplementary Table 2, and the corresponding photo of fractured samples is presented in Supplementary Fig. 16, Supplementary information.

9. Ionic liquids are usually expensive, so their recyclability is crucial. After bonding the wood, can the ionic liquid in the cellulose solution be recovered and reused?

Response: Yes, the ionic liquid used for wood bonding is recoverable and reusable. To evaluate the reusability of [emim][OAc] in our process, we recovered [emim][OAc] from the wastewater collected during the water regeneration (Supplementary Fig. 25c). The wastewater containing mainly [emim][OAc] and water was first vacuum filtrated through a membrane filter (Polycarbonate, pore size: 0.1 μm) to remove insoluble fractions (Supplementary Fig. 25a). The purified [emim][OAc]/water mixture was then concentrated via rotary evaporation (Supplementary Fig. 25b). To further eliminate residual water, the concentrated solution was then oven-dried at 105 °C overnight. The recovered [emim][OAc] demonstrates a color change from yellowish to brownish; however, no noticeable changes in its chemical structure were detected by FTIR (Supplementary Fig. 25d), except for slight increase in -OH band intensity at high wavenumber, possibly due to difficult-to-remove residue water. Subsequently, 5 wt.% pulp was dissolved using this recovered [emim][OAc] to prepare a cellulose-IL solution. Wood substrates were bonded using this solution (Supplementary Fig. 25e), and successfully bonding was achieved. The bonded wood samples exhibited a shear strength of 13.7 MPa (Supplementary Fig. 26), demonstrating the effective reusability of the recovered [emim][OAc]. The reduced shear strength might be due to reduced cellulose solubility in the presence of aging products of [emim][OAc], such as *N*-methylimidazole and imidazole, which share similar structure as [emim][OAc] and thus can hardly be identified by FTIR (see attached reference).

Ref: Yoneda, Y., Hettegger, H., Böhmendorfer, S. *et al.* Cellulose acylation in aged 1-ethyl-3-methyl-imidazolium carboxylate ionic liquids upon fiber spinning. *Cellulose* (2025). <https://doi.org/10.1007/s10570-025-06583-y>

Supplementary Fig. 25. Purification, recovery, and re-use of IL [emim][OAc]. a) purification setup for removing insoluble fraction from [emim][OAc]-containing wastewater. b) rotary evaporation process for removing water from the purified [emim][OAc]/water mixture. c) photo showing the original [emim][OAc], collected wastewater, purified wastewater, and recovered [emim][OAc]. d) FTIR spectra comparing the original [emim][OAc] and recovered [emim][OAc]. e) photo of a 5 wt.% pulp solution dissolved in the recovered [emim][OAc] and the resulting wood assemblies.

Supplementary Fig. 26. Shear strength of wood bonded with 5 wt.% pulp solution prepared using recovered [emim][OAc], along with corresponding photo of the fractured specimens.

10. The viscosity, dissolution capability, and stability of different ionic liquids may significantly impact bonding performance. This may require further elaboration on the criteria used for selecting the ionic liquids.

Response: Yes, viscosity and dissolution capability may influence bonding performance. The ability of an IL to dissolve cellulose plays a critical role in forming a homogenous solution and enabling effective interaction with wood adherents. In the context of wood bonding process, using an IL that remains liquid at room temperature (20 °C) enhances the practical applicability of the bonding agent to wood adherents. In this study, we selected [emim][OAc] for cellulose dissolution, due to its low melting temperature, high cellulose solubility, low toxicity, and large industrial availability. We have added the following clarification in the revised manuscript (Page 3, Lines 19-20):

...This approach takes advantage of the remarkable solubility of both cellulose and native wood cell wall in the room-temperature ionic liquid (IL)—1-Ethyl-3-methylimidazolium acetate [emim][OAc]...

11. The numbering of figures in the manuscript contains errors and should be corrected. For example, on page 13, line 23, the figure reference should be corrected to "Fig. S9."

Response: We have revised all the figure reference to ensure the readability of our revised manuscript.

12. In Figure 5, the material names described in the main text are not consistent with those shown in the figure legend and need to be corrected.

Response: We have revised the material names to assure the consistency.

Reviewer #3 (Remarks to the Author):

Zhang and coworkers have submitted a very interesting and well-elaborated manuscript on a new concept to achieve strong wood bonding without using petroleum-based adhesives. The topic is timely and highly relevant because petroleum-based adhesives strongly contribute to the carbon footprint of wood-based products and are one of the decisive factors for the rather low recycling rate of wood. The achieved properties are extremely impressive, in particular the wet strength properties, which are usually the weak spot of alternative wood gluing systems. The comprehensive characterization provides important information on structural and chemical features as well as the resulting mechanical properties. However, these findings can only partly explain the obtained mechanical performance.

Response: We thank the reviewer for their interest in our work and for recognizing its significance. We have obtained additional result to further explain the high mechanical performance achieved in our study.

Specific comments:

1. Page 2, line 4: Better “carbon storage” than “carbon sink”, since sink refers to a different time frame.

Response: ‘Carbon sink’ has been changed to ‘carbon storage’ in the revised manuscript.

2. Page 2, lines 10-11: I think the argument of limited bond strength of petroleum-based adhesives is questionable since several of these products are established in the timber engineering industry.

Response: The claim of ‘limited bonding strength’ has been removed.

3. Page 3, line 9: I am wondering whether friction welding is an appropriate reference system for this approach.

Response: We have removed the statement suggesting a resemblance between the two bonding approaches. However, as friction welding represents one of the key novel wood bonding methods, the discussion of the bonding line structure formed during friction welding remains relevant in the introduction and has been retained.

4. Page 4, line 15: the authors claim scalability and implementation potential for the wood industry. Therefore, they should discuss the press conditions in terms of temperature, pressure,

and duration. Related to implementation, the relevance of less perfect surfaces should be addressed, in particular the influence of knots.

Response: We conducted bonding tests on samples prepared with pulp-IL solution at varying pulp concentrations and regenerated using different methods, as well as subjected to hot pressing at different temperatures and pressures. In addition, wood substrates that contain knots were also bonded. These experiments were designed to provide a more comprehensive evaluation of the bonding strategy. The results of these bonding tests, along with corresponding photos of fractured samples, have been included in Supplementary Information. In summary, the current bonding protocol consistently yielded the best performance. A comprehensive discussion of the results was added to Supplementary Information as Supplementary Notes.

As shown in Supplementary Fig. 22, wood assemblies subjected to 10 min hot-pressing achieved a bonding strength of 17.5 ± 1.2 MPa, while those pressed for 60 min yielded 19.1 ± 0.6 MPa. The 10 min hot pressed samples show slightly lower strength compared to those pressed for 30 min (19.6 ± 1.3 MPa), whereas the 60 min hot-pressing results were comparable to the 30 min samples. These observations indicate that extending the hot-pressing duration up to 30 min can enhance bonding performance, likely due to improved penetration of cellulose solution into the cell lumina and formation of a more integrated bonding interface.

From an industrial perspective, however, a 10 min hot-pressing duration is sufficient to produce bonded wood with good bonding strength, balancing performance with process efficiency.

Supplementary Fig. 22. The bonding strengths and the fracture images of wood samples bonded with 5 wt.% pulp-IL solution and hot-pressed for different durations at 140 °C with 1.5 MPa pressure. a) wood sample hot-pressed for 10 min. b) wood sample hot-pressed for 60 min.

As shown in Supplementary Fig. 23, wood bonding was performed using a 5 wt.% pulp-IL solution at 140 °C under reduced hot-pressing pressure of 0.1 and 1 MPa. Notably, wood samples bonded under 0.1 MPa pressure exhibited complete delamination after water regeneration, indicating poor interfacial adhesion. In contrast, samples bonded under 1 MPa pressure did not show delamination but they demonstrated reduced bonding strengths of 14.3 ± 0.7 MPa compared to those bonded at 1.5 MPa (19.6 ± 1.3 MPa). These results suggest that effective wood bonding requires sufficient pressure to promote cellulose penetration and activate the softening effect of ILs on cell wall components at the bonding interface.

Supplementary Fig. 23. The bonding strengths and the images of fractured wood samples bonded with 5 wt.% pulp-IL solution and hot-pressed at different pressures for 30 min. a) wood sample hot-pressed at 140 °C with 1 MPa pressure. b) wood sample hot-pressed at 140 °C with 0.1 MPa pressure.

In response to the concern regarding the implementation of less ideal wood substrates in industrial applications, wood specimens containing knots were bonded using a 5 wt.% pulp-IL solution (Supplementary Fig. 24a). The corresponding shear strength was evaluated, and both the measured values and the image of fractured samples are demonstrated in Supplementary Fig. 24b. As shown in Supplementary Fig. 24a, the wood substrates containing knots were successfully bonded; notably, no visible delamination occurred even after water washing. The shear strength of the samples containing knots (9.8 ± 2.6 MPa) was lower than that of the defect-free samples (19.6 ± 1.3 MPa), likely due to the structural discontinuities introduced by the knots. Nevertheless, these results suggest that while knots may reduce the overall shear

strength, they do not inhabit the bonding capability of the proposed method. Moreover, the shear strength remains within a range that is comparable to typical values for the substrates, indicating the potential suitability of this bonding approach for industrial applications involving less ideal wood materials.

Supplementary Fig. 24. Photos of wood substrates containing knots and the corresponding bonding strength of samples. a) photo of wood substrates containing knots before and after bonding with 5 wt.% pulp-IL solution and hot-pressed at 140 °C with 1.5 MPa pressure for 30 min, and subsequent regeneration in water. b) the bonding strength and the image of fractured wood samples containing knots

5. Page 5, lines 2-4: “Notably, the cell wall of both adherends intertwined, forming a mechanically interlocking structure resembling that of friction-welded wood.” I am skeptical about this statement since the loading cases are entirely different. The authors may better explain the nature of the intertwined structures.

Response: We thank the reviewer for the comment. To avoid confusion due to the difference in loading conditions, we have removed the statement comparing the cellulose-IL bonded wood to friction-welded wood.

6. Page 5, lines 17-18: it should be added that staining does not allow for a quantification.

Response: We have revised the manuscript as follow (Page 5, Line 19):

...Wiesner staining can visualize the **qualitative** structural changes of lignin...

7. Page 6, lines 20-21: I would expect earlywood cell walls to be thinner. Did you measure individual cell walls or double cell walls (lumen to lumen)?

Response: We measured the compound cell wall thickness (lumen to lumen) and have clarified this in the revised manuscript.

8. Page 7, lines 5-11: I am very surprised that the authors did not detect cellulose II. The authors should conduct an XRD analysis with a sample cut into small pieces to eliminate the structural orientation pattern.

Response: We understand the reviewer's concern regarding the detection of cellulose II. Our WAXS measurements were conducted using a bench beamline with an X-ray beam size of $400\ \mu\text{m} \times 400\ \mu\text{m}$ (shown in **Fig. R6**). This beam size was sufficient to fully cover the bonding interface (approximately $500\ \mu\text{m}$ wide) ensuring that the resulting XRD data specifically reflect the structural features at the interface. The regeneration of cellulose dissolved in ionic liquid doesn't always result in cellulose II. Samayam *et al.* has reported that, upon washing and drying, cellulose dissolved by ionic liquid could recrystallize to either cellulose I β or cellulose II depending on the severity of the IL treatment. Ling *et al.* also reported that the recrystallization behavior and generation of intermediate phases may prohibit the converting to cellulose II allomorph from cellulose I.

Reference:

I. P. Samayam, B. L. Hanson, P. Langan, C. A. Schall, Ionic-liquid induced changes in cellulose structure associated with enhanced biomass hydrolysis. *Biomacromolecules* 12, 3091-3098 (2011). doi: 10.1021/bm200736a

Z. Ling, S. Chen, X. Zhang, K. Takabe, F. Xu, Unraveling variations of crystalline cellulose induced by ionic liquid and their effects on enzymatic hydrolysis. *Sci. Rep.* 7, 10230 (2017). doi: 10.1038/s41598-017-09885-9.

Fig. R6. Photo of a wood specimen mounted on the WAXS sample holder. Yellow squares indicate the regions where WAXS measurements were conducted.

9. Page 9, lines 14-16: A penetration through pits is known for other adhesives as well. A peculiar feature of many formaldehyde-based adhesives is their ability to penetrate the cell wall.

Response: We have revised the manuscript as follows (Page 9, lines 21-23):

...It was evident from the color maps that the regenerated cationic cellulose predominantly filled the lumina of earlywood tracheids at the bonding interfaces of earlywood-to-earlywood and earlywood-to-latewood. **Its presence in the cell wall was also detected, though less pronounced than in the lumina.** This penetration behavior resembles that of formaldehyde-based adhesives **as well as other low viscosity adhesives**, which typically flow through pits between tracheid or fiber cells to form a network of adhesive polymers enabling efficient bonding...

10. Page 13, lines 2-6: The used pine species specimens possess large, window-like, cross-field pits, which may also play an important role in the transverse penetration processes. This should be tested by doing a control experiment with a species like spruce.

Response: As suggested by the reviewer, bonding tests were conducted using spruce specimens. The spruce specimens were successfully bonded, and the bonding strength is presented below and as Supplementary Fig. 11. As expected, the bonding strength of spruce (14.0 ± 0.9 MPa) was lower than that of Pine (19.6 ± 1.3 MPa), which can be attributed to the

lower density (320 kg m^{-3}) as well as the lack of large, window-like, cross-field pits. Nevertheless, the shear strength measured for spruce bonded with pulp-IL remains higher than that of solid spruce (5-10 MPa).

Additionally, fluorescence microscopy revealed that the cell lumina at the bonding interface in spruce were filled with regenerated cellulose, demonstrating a similar microstructure to that observed in pine wood (Supplementary Fig. 12).

Supplementary Fig. 11. The shear strength and the photo of fractured spruce samples bonded with 5 wt.% pulp-IL solution.

Supplementary Fig. 12. Fluorescence microscopy images showing calcofluor white stained cross-sections of wood bonded using 5 wt.% pulp-IL solution

11. Page 13, lines 17-19: I find this statement on the “interconnection” ability a bit vague. What is the actual mechanism? What is the type of interaction between the cell wall and the added cellulose?

Response: Interconnection was likely achieved by the continuous cellulose network formed by regeneration. The interaction between the cell wall and the added cellulose was mainly facilitated by physical entanglement of polymers during dissolution and regeneration process. We have revised the manuscript as follows (Page 14, lines 3-4):

...Indeed, the results demonstrate that higher molecular weight cellulose enables the formation a regenerated cellulose matrix inside the cells **with possible physical entanglement with the cell wall polymers**, effectively interconnecting adjacent tracheids at the bonding interface.

12. Page 14, lines 9-13 and 16-18: The achieved shear strength is very remarkable, but it appears questionable, how it can be that high. If a failure appears in the bulk (and with a certain distance from the bondline), the shear strength of the wood is the determining factor. In the test setup, the shear strength of pine wood should be around 10 MPa. Even if one considers a certain level of densification these values are still far off. The authors need to provide a more elaborated interpretation of how such high values can be obtained, which goes beyond the rather vague “multiscale mechanism” addressed on Page 15, lines 7-12.

Response: We have optimized the sample geometry for tensile shear test according to Reviewer #1’s comment. After adopting a standard 10 mm overlap length, we have obtained updated shear strength for wood bonded with different types of cellulose: MCC (15.7 ± 2.7), MFC (17.2 ± 0.9 MPa), and Pulp (19.6 ± 1.3 MPa). Although the measured values decreased compared to our previous results, still exhibited significantly higher shear strength than the solid wood. Wood bonded with MCC-IL, which lacks lumen-filling structure, showed only slight increase over solid wood. In contrast, wood bonded with pulp-IL featuring lumen filling structure showed substantially higher shear strength.

These results confirmed our interpretation that high shear strength was achieved by the formation of a unique structure consisting of 1) a densified layer enabling mechanical interlocking of cells; 2) a network of regenerated high molecular weight cellulose that not only fills the lumen but also interconnects adjacent tracheids; 3) potential physical entanglement between the added cellulose and native cell wall polymers.

13. Page 16, lines 1-6: The obtained wet shear strength is extremely impressive. Again, I see the problem that these values are far above the shear strength of wet wood. Additionally, the in-depth characterization of the bond line does not provide sufficient evidence for such extraordinary water resistance. Do we miss an important piece of the puzzle? Tensile-shear tests are a good performance indicator but for the claimed implementation potential delamination tests in drying and wetting cycles are by far more relevant. These tests should be conducted.

Response: The bonding strength reported in our study refer to the dry bonding strength of the wood samples. After regeneration with water, the bonded samples were air-dried under ambient conditions (20 °C and 30% RH). This clarification has been added to the Bonding of wood blocks and plywood production section in the Materials and Methods (Supporting information Page 4).

The delamination test, i.e. boiling test, was conducted following the European Standard EN 15425:2023 Annex B. The bonded wood specimens were boiled at 100 °C for 6 h and followed by water soaking at 20 °C for 2 h. After test, no delamination was observed (Supplementary Fig. 13). the excellent water resistance of the bonding line is attributed to the permanent cell wall deformation induced by partial dissolution with the ionic liquid and subsequent regeneration during water exposure. In our previous work, we demonstrated that ionic liquid treatment alone can effectively suppress the set-recovery of densified wood. No swelling of cell walls was observed even after 24 hours of water immersion (see Fig. 4 in the reference).

Reference:

Zhang, S., Meinhard, H., Collins, S. *et al.* Effect of ionic liquid [emim][OAc] on the set recovery behavior of densified wood. *Cellulose* **31**, 8267–8278 (2024).

<https://doi.org/10.1007/s10570-024-06043-z>

Supplementary Fig. 13. Pulp-bonded wood subjected to delamination test. Bonded wood was boiled at 100 °C for 6 h and followed by 2 h of water soaking at 20 °C.

Reviewer #1 (Remarks to the Author):

The authors have provided a comprehensive and well-structured response to the concerns raised in the previous review. In particular, the additional data and analyses presented on the mechanical performance and adhesion mechanisms significantly strengthen the scientific rigor and clarity of the manuscript. The newly added mechanical characterization is both quantitative and comparative, offering a more robust validation of the material properties. Furthermore, the improved discussion and supporting evidence regarding the adhesion mechanisms enhance the mechanistic understanding and provide greater persuasive power to the overall conclusions. At this stage, I have no further concerns. I believe the manuscript has been substantially improved and now meets the standards for publication in Nature Communications. I recommend acceptance.

Response: We sincerely thank the reviewer for the positive feedback and for recognising the improvements made to the manuscript.

Reviewer #2 (Remarks to the Author):

The authors have satisfactorily addressed all the points I raised during the review. I recommend the manuscript for publication in its current form.

Response: We thank the reviewer for the encouraging comments and are glad that the revisions have addressed all concerns.

Reviewer #3 (Remarks to the Author):

The authors conducted additional experiments and have convincingly responded to the raised concerns. No further comments.

Response: We appreciate the reviewer's positive assessment and are pleased that the additional experiments were found convincing.

The manuscript presents an innovative strategy to bond wood using a solution of cellulose in an ionic liquid. This approach is interesting and has potential implications for sustainable material development. However, there are significant issues with the experimental design of the mechanical performance tests, which compromise the validity of the results regarding the adhesive's bonding strength. Without addressing these critical issues, the discussion of the bonding strength of the adhesive remains unsubstantiated. I recommend a major revision. Only after resolving the following issues can the manuscript be reconsidered for publication.

1. In all tensile shear tests conducted, failure occurred in the bulk wood rather than at the bonding interface. This indicates that the experimental setup does not accurately reflect the adhesive's shear strength. Specifically, the maximum force corresponding to reported approximately 23 MPa shear stress still reflects fracture force of the bulk wood, which is approximately twice the tensile strength of wood. This outcome is expected based on the test specimen design illustrated in Fig. S9, where the tensile area of the wood is doubled while the bonded area matches the cross-sectional area of a single wood specimen.

While the data demonstrate that the adhesive likely exhibits a shear strength exceeding the intrinsic tensile strength of wood, the absolute values in Fig. 5b are not comparable across different experimental groups. Consequently, the performance of various types of cellulose in influencing the adhesive's properties cannot be reliably assessed. Furthermore, any interpretation of Fig. 5, as well as its correlation with Fig. 4, must be revisited once the accurate shear strength is measured.

I am unsure whether the authors referenced any ISO or ASTM standards for their testing. However, the experimental design could be modified to better align with the adhesive's material properties. For example, reducing the bonded area would ensure that the bonding interface fails under lower tensile forces, avoiding bulk wood fracture and yielding data more representative of the adhesive's true performance.

2. The manuscript inconsistently refers to the adhesive as "cellulose-IL" and "pulp-IL," for example, in lines 18 and 20 on page 6. Please standardize terminology or provide clear explanation.

3. In Supplementary Table 1 and Fig. S3, the description of earlywood cell wall swelling in ionic liquid is based on measurements of five locations on

the cell wall from the same sample. However, for the results to hold statistical significance and represent general trends, the selected cell wall measurements should ideally be derived from multiple samples and multiple cell walls. Relying on a single sample limits the robustness of the data and may introduce sampling bias.

4. The manuscript employs two distinct approaches to investigate the distribution of cellulose within wood: cationic modification of cellulose in Fig. 3 and characteristic staining of cellulose in Fig. 4. However, the rationale for using different methods in these experiments is unclear. The authors should clarify why a staining-based method was not employed in Fig. 3, as it appears to be a more direct and less disruptive approach for visualizing cellulose distribution. The use of cationic-modified cellulose could interfere with cellulose-lignin interactions, potentially altering the natural distribution of cellulose. Additionally, the delignification process may have affected the cellulose structure, introducing further uncertainties. These factors need to be carefully considered and discussed to validate the results presented in Fig. 3.

5. I recommend a deeper investigation into the interaction mechanisms between cellulose in ionic liquid (IL) and wood. Intuitively, this interaction depends on the diffusion of cellulose during the hot-pressing process and the phase separation induced by water. While the authors have explored the influence of molecular weight on cellulose distribution, primarily in the context of the diffusion process, the impact of this factor on mechanical performance requires reexamination and validation.

Beyond molecular weight, it would be favorable to investigate additional factors that may significantly contribute to the adhesive performance. What I can think of includes the concentration of cellulose in IL and the kinetics of regeneration induced by water, which might influence the bonding behavior. These parameters could play critical roles in determining the distribution, interaction, and eventual performance of the adhesive. If the authors could systematically elucidate how such factors affect the cellulose-wood interaction and establish a correlation with the mechanical results, they would provide a more comprehensive and robust mechanism for this bonding strategy. Such insights would greatly enhance the scientific significance of the study.